# FLEXIBLE FEATURE DISTILLATION FOR LARGE LANGUAGE MODELS

## ABSTRACT

Knowledge distillation (KD) has become a cornerstone for compressing large language models (LLMs). However, existing LLM-KD methods have primarily focused on logit-based approaches, which achieve good performance but overlook the rich internal representations of LLMs. Feature-level KD could leverage this structure to provide complementary benefits, yet it remains underexplored because current feature-KD approaches typically assume identical teacher–student hidden sizes, a restrictive and unrealistic assumption. A common workaround is to train a linear projector to align their feature spaces; however, this introduces additional parameters, distorts teacher embeddings, and often degrades downstream performance, especially in generative tasks. We propose Flex-KD, a parameter-free framework for task-driven feature distillation for LLMs. Instead of projecting the entire teacher representation, Flex-KD uses gradient-based scores to identify the most task-relevant dimensions of the teacher's hidden states and distills only this subspace into the student. This ensures that the student's limited capacity is allocated to informative components, while avoiding projector-induced distortion and extra parameters. Flex-KD integrates seamlessly with existing KD pipelines and supports differing teacher–student hidden sizes. Extensive experiments across both classification and generative tasks, i.e., instruction-following and summarization, show that Flex-KD **consistently** boosts the student performance, achieving up to a 3.75% performance gain over the linear projection baseline.

## 1 INTRODUCTION

Recently, there has been a surge in using large language models (LLMs) for classification (Liang et al., 2021; Jiao et al., 2020; Sanh et al., 2019) and generative tasks (Liu et al., 2024; OpenAI, 2023; Team et al., 2023), where they achieved strong performance across diverse applications (Zhuge et al., 2024; OpenAI, 2023; Touvron et al., 2023; Wang et al., 2023b). Despite their remarkable success, these models are computationally intensive and often impractical for deployment in resource-constrained environments. Hence, there has been an interest in making LLMs more efficient in terms of storage and computation through knowledge distillation (KD) (Hinton, 2015; Zhu et al., 2024; Xu & McAuley, 2022). KD can be applied during pre-training to create general-purpose compressed models (Jiao et al., 2020; Sanh et al., 2019; Liu et al., 2024), or more efficiently, motivated by findings from Kovaleva et al. (2019), KD can be applied during fine-tuning to produce task-specific distilled models (Zhou et al., 2021; Liang et al., 2020; Sun et al., 2019a; Gu et al., 2024a; Ko et al., 2024b).

In the era of LLMs, most KD research has focused on logit distillation (Ko et al., 2024b; Gu et al., 2024a; Taori et al., 2023; Kim & Rush, 2016), i.e., transferring the output probabilities (soft labels) of the teacher model to the student. In contrast, feature distillation (Sanh et al., 2019), which transfers intermediate hidden representations from teacher to student, has received far less attention, even though it has demonstrated strong results in classification tasks (Sun et al., 2019b; Dasgupta & Cohn, 2025; Saadi et al., 2023b). This disparity can be attributed to a fundamental limitation of conventional feature distillation methods: the requirement that teacher and student models have identical hidden dimensionalities (Muralidharan et al., 2024; Sun et al., 2019a; Sanh et al., 2019), which considerably restricts their applicability across diverse architectures. A common solution is to introduce a learnable linear projector to match the student's feature representation with that of the teacher (Chen et al., 2022; Jiao et al., 2020). While this approach has proven effective in removing the constraint (Miles & Mikolajczyk, 2024; Chen et al., 2022; Jiao et al., 2020), it adds extra parameters that must be

trained from scratch during the fine-tuning distillation process and may distort the teacher's feature representations. This can harm student performance, particularly in downstream distillation scenarios where training data is limited (Dasgupta & Cohn, 2025).

In this paper, we propose a novel task-specific KD method, Flex-KD, which enables effective hidden state matching between teacher and student models with differing hidden sizes, without introducing any additional parameters. The key intuition behind Flex-KD is that LLMs are over-parameterized for domain-specific tasks and that only a subset of their units contributes significantly to a given task (Hase et al., 2024; Kovaleva et al., 2019). Flex-KD focuses on identifying these task-relevant units and distilling knowledge only from their subspace to the student model.

In the standard task-based KD framework for LLMs, the teacher is typically a large, versatile model (Gu et al., 2024a), pre-trained on diverse datasets (Wang et al., 2023a; OpenAI, 2023). Distillation often transfers all components of the teacher uniformly to the student (Gu et al., 2024a; Peng et al., 2023; Kim & Rush, 2016; Sanh et al., 2019), a strategy that would help if the goal was to train generally capable student models (Jiao et al., 2020; Sanh et al., 2019). However, many real-world applications prioritize performance on specific downstream tasks (Ge et al., 2023), where transferring the full versatility of the teacher may introduce unnecessary complexity.

In fact, recent studies (Hase et al., 2024; Gromov et al., 2024; Luo et al., 2024; Dai et al., 2021) show that only a subset of LLM components significantly contribute to task performance. To further confirm this phenomenon, in Figure 1, we visualize the activations of the last hidden layer on a downstream task example: many units display near-zero or low-magnitude activations, suggesting limited contribution to the final output. This indicates that indeed distilling all hidden units from the teacher is not only unnecessary but may even hinder specialization. Additional visualizations with varying thresholds are provided in Appendix A. Our proposed approach Flex-KD is designed to leverage these findings and, rather than relying on uniform transfer or rigid projector-based mappings, effective task-based distillation should be selective.

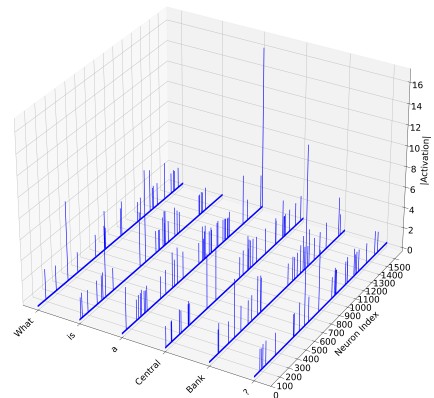

Figure 1: Last-layer activation magnitudes (z-axis) of a fine-tuned GPT-xlarge on a downstream example, with values < 2 set to zero. The x/y axes denote sequence and features.

Flex-KD is a novel task-driven distillation method that enables hidden state matching between teacher and student models with differing hidden sizes, without introducing any additional parameters. Specifically, given a student with hidden size $d_S << d_T$ (the teacher model with size $d_T$), for a given task, our method first assigns task-conditioned importance scores to different units in the teacher hidden layer. These nodes are then ranked by their importance, and the subspace formed by top $d_S$ units are selected and prioritized for distillation. This forces the student model to concentrate its limited capacity on the most relevant teacher components, thereby improving performance while accommodating flexible hidden layer sizes.

In summary, our contributions are as follows:

- We propose **Flex-KD**, a novel task-driven feature distillation method that enables effective knowledge transfer between teacher and student models with differing hidden sizes, consistently outperforming existing baselines while supporting flexible student architectures.

- Unlike existing methods, we design Flex-KD to be selective and parameter-free, enabling faithful transfer of task-relevant knowledge from teacher embeddings. Moreover, it can be seamlessly integrated with existing logit-based distillation methods to further enhance performance.

- Extensive experiments across various language generation (seven datasets and six models) and classification (six datasets and two models) benchmarks demonstrate that Flex-KD **consistently** outperforms state-of-the-art baselines. Specifically, we achieve performance gains of up to 1.79% on classification, 2.1% on instruction-following, and up to 3.75% on summarization compared to the standard linear projection approach.

## 2 RELATED WORK

**Knowledge distillation** (Schmidhuber, 1992; Hinton, 2015) is a widely used model compression technique that transfers knowledge from a large teacher model to a small, efficient student model (Sanh et al., 2019; Gou et al., 2021). In natural language processing (NLP), KD has been predominantly applied to text classification tasks by aligning the student model with the teacher's output distributions (Liang et al., 2021; Zhang et al., 2023b), hidden representations (Sun et al., 2019b; Jiao et al., 2020), or attention matrices (Wang et al., 2020; 2021). These approaches effectively reduce model size while preserving performance, making them suitable for resource-constrained setups. However, the application of KD in language generation tasks is more complex than in classification tasks (Gu et al., 2024a). Unlike the fixed-label space of classification, open-ended text generation involves producing discrete token sequences of varying lengths, which adds inherent complexity.

**Logit distillation** (Hinton, 2015) aims to minimize the distance between student and teacher output distributions. Current KD techniques for generative models are mainly centered around logit-based methods, where they primarily minimize the forward Kullback-Leibler divergence (FKLD) (Kullback, 1951) between the teacher and student model output distributions (Sanh et al., 2019; Kim et al., 2024b). This may involve supervision using the teacher's outputs at each generation step (Kim & Rush, 2016; Taori et al., 2023), training on teacher-generated text (Peng et al., 2023), or employing reverse Kullback-Leibler divergence (RKLD)(Gu et al., 2024a; Kim et al., 2024a; Gu et al., 2024b), which makes the student distribution focus on certain modes in the teacher's distribution. Recent work (Wang et al., 2025; Ko et al., 2024a) has found that the performance difference of FKLD and RKLD closely depends on the dataset and the task at hand.

**Feature distillation** (Muralidharan et al., 2024; Jiao et al., 2020) has received less attention in generative tasks (Muralidharan et al., 2024) compared to logit-based methods, which can be explained by the inherent limitation of conventional feature KD approaches that enforce equal hidden dimensionalities between teachers and students. This restriction reduces both student architectural flexibility and compressibility. A common workaround, adapted from vision and classification (Chen et al., 2022; Jiao et al., 2020), is to train an additional linear projector to align the teacher's and student's feature spaces (Jiao et al., 2020). While effective in pre-training (Jiao et al., 2020), projectors often under-perform in downstream tasks (Dasgupta & Cohn, 2025), where data is scarce, introduce extra parameters, and may distort teacher features. The work closest to ours is Dasgupta & Cohn (2025), which also tackles the problem of feature distillation between teacher and student with differing hidden sizes. Their approach introduces a metric to compute similarity between tensors of mismatched dimensions, enabling flexible hidden state distillation. However, it still uniformly transfers knowledge from all teacher components without considering their task relevance. As shown in Table 4, this limitation can degrade student performance in several cases. This motivates the need for filtering out non-relevant units and focusing on the task-relevant subspace, the core idea of our proposed approach.

## 3 FLEX-KD

In this section, we introduce Flex-KD, which enables feature KD between teacher and student models with different hidden sizes. Figure 2 illustrates the Flex-KD workflow: a teacher model, fine-tuned on the downstream task $t$, provides hidden representations $h^T$ that capture task-relevant features. Our goal is to distill only the teacher dimensions that matter for task $t$. Concretely, we minimize the student-teacher cross-correlation on a selected teacher subspace (Eq 5). To obtain that subspace we compute gradient-based importance scores (Eq 1- 4) and select the top units. Specifically, we compute the gradient of the output with respect to each unit's activation. The gradient magnitude reflects each unit's influence on the output, serving as a task relevance score (Krishna et al., 2024; Simonyan et al., 2014). Units with higher scores are prioritized for distillation. Subsections 3.1 and 3.2 describe our approach for selecting the top task-relevant units and detail the distillation framework, respectively.

### 3.1 TASK-RELEVANT UNITS LOCALIZATION

In this subsection, we describe our method for identifying task-relevant units in the teacher model. Formally, let the teacher model $T$ be a neural network $F$ with hidden size $d_T$, and let the student model $S$ be a neural network with hidden size $d_S$, where $d_S << d_T$. The teacher model $T$ is fine-tuned on dataset $D = \{x_1, x_2, \ldots, x_N\}$ to perform task $t$. Focusing on the last hidden layer, we assume that the hidden states of the teacher and student networks are $h^T \in \mathbb{R}^{d_T}$ and $h^S \in \mathbb{R}^{d_S}$,

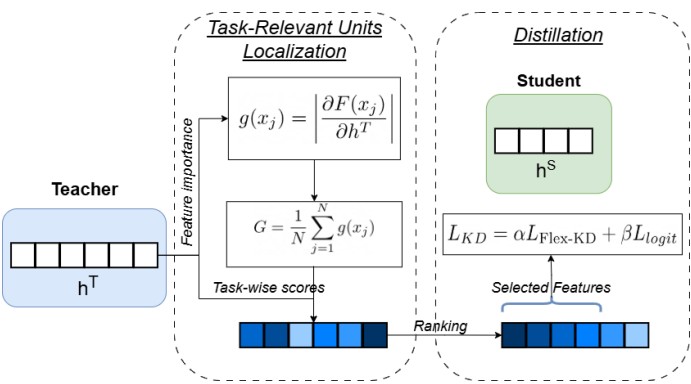

Figure 2: Overview of Flex-KD.

respectively. In our approach, we use the gradient $\frac{\partial F(x_j)}{\partial h^T}$ to quantify the influence of each component of the hidden state $h^T$ on the model's output for a given input $x_j \in D$. Intuitively, the gradient allows us to capture how small variations in the hidden representation $h^T$ affect the model's prediction $F(x_j)$. A unit output (i.e., a dimension in $h^T$) with a large gradient magnitude is considered highly sensitive, indicating that it plays an important role in determining the model's output for the sample $x_j$. Hence, such a unit is prioritized during distillation.

Formally, the importance scores of all the hidden units in the hidden state $h^T$ for the input sample $x_j$ are computed as follows:

$$g(x_j) = \left| \frac{\partial F(x_j)}{\partial h^T} \right| \in \mathbb{R}^{d_T}, \tag{1}$$

where $|\cdot|$ denotes the element-wise absolute value. Here, entry $g(x_j)_i$ represents the importance score of the $i$-th unit to perform prediction on the input $x_j$.

**Importance Scores.** In the context of LLMs, recent work have highlighted the effectiveness of gradient-based methods, i.e., the derivatives of the loss or output with respect to weights, masks, or activations, for identifying key network components (e.g., units), in contrast to gradient-free approaches such as magnitude-based metrics (Iurada et al., 2025; Guo et al., 2025; Ma et al., 2023; Fu et al., 2022; Yu et al., 2022; Liu et al., 2021). More advanced gradient techniques, such as Integrated Gradients (IG) (Sundararajan et al., 2017), provide more robust attribution signals but come at significant computational cost. Standard gradient methods (Nielsen et al., 2022; Ding & Koehn, 2021; Li et al., 2016), by requiring only a single backpropagation pass, offer a favorable balance between efficiency and accuracy. While it can suffer from saturation in deep networks, this issue is largely mitigated when analyzing the gradient of the output with respect to the final layer, as done in our proposed Flex-KD (Subsection 3.2). The final layer directly reflects model predictions, offering an efficient and reliable importance signal. To further motivate our choice, in Subsection 4.1 Figure 3a, we conduct an experiment to investigate different selection strategies, showing that standard gradients outperform both activation magnitudes and IG, achieving higher performance with lower variance, thereby confirming its stability. Beyond comparing attribution strategies, we also assess whether gradient-based importance correlates with the *actual* contribution of units to task performance. To this end, we conduct a neuron-removal study on the fine-tuned teacher across three GLUE tasks (RTE, STS-B, SST-2), summarized in Table 9 (Appendix). For each task, we progressively remove a given fraction of units in the final hidden layer according to three criteria: (i) highest gradient-based importance, (ii) lowest gradient-based importance, and (iii) random selection. We observe that removing high-importance units leads to substantial degradation (e.g., up to $-13.7$ accuracy points on RTE and $-6.5$ points on STS-B), whereas removing the lowest-ranked units has almost no effect, and random ablations cause only mild or moderate drops. This consistent gap between high- and low-importance subsets provides direct evidence that gradient magnitudes capture task-critical neurons, reinforcing our choice to use them as the basis for selecting the teacher subspace distilled by Flex-KD.

In our method, to compute the overall importance scores for task $t$, we aggregate the results over all $N$ samples in $D$ as follows:

$$G = \frac{1}{N} \sum_{j=1}^{N} g(x_j) \in \mathbb{R}^{d_T}, \tag{2}$$

where $G_i$ represents the importance score of the $i$-th unit in the teacher hidden layer to perform task $t$. Since the importance scores of all $d_T$ teacher units have been computed for task $t$, we select the top $d_S$ most relevant units to match the hidden size of the student model ($d_S$ is the student hidden size ).

We achieve that by ranking the units based on their obtained importance scores as follows:

$$R = \left\{ G_{i_1}, \ldots, G_{i_{d_T}} \mid G_{i_1} \geq \cdots \geq G_{i_{d_T}} \right\}, \tag{3}$$

where $\{i_1, \ldots, i_{d_T}\} \in [d_T]$ and $R$ correspond to the set of rearranged importance scores so that the highest score is the first in the set and the lowest score is the last in the set. Thus, the top task-relevant units are those corresponding to the top scores in $R$. In particular, the ranked set $R$ enables us to select flexible hidden representations from the teacher model that capture the most task-relevant knowledge. For instance, $h^T_{i_1:i_8}$, where $i_8 \ll i_{d_T}$ denotes the output feature representations of the top-8 relevant units for task $t$. Since the student model has a hidden size of $d_S$, we select the top-$d_S$ units from the teacher model corresponding to the highest $d_S$ scores in the ranked set $R$, and use their corresponding subspace as distillation target. The top-$d_S$ task-relevant units from the teacher are:

$$E = \{i_1, i_2, \ldots, i_{d_S}\}, \tag{4}$$

where each $i_k$ refers to the index of the k-th selected unit in the given teacher hidden layer. This carefully selected set $E$ allows us to perform *flexible hidden state matching* between student and teacher models with different hidden dimensions $d_S$ and $d_T$, respectively. To sum up, Flex-KD enables flexible and effective feature distillation by directing the student's limited capacity toward the most relevant teacher components, while disregarding less informative or irrelevant ones.

## 3.2 Distillation

As we have identified and selected the top-$d_S$ task-relevant units from the teacher model, in this subsection, we describe the distillation process. To transfer the knowledge of the subspace of the carefully selected set of units to the student model $S$, we employ a correlation loss function, which was shown to be more effective than traditional mean squared error (MSE) and cosine distance in capturing meaningful relationships in the feature space (Saadi et al., 2023a; Fard & Mahoor, 2022). For completeness, our results in Table 7 in Appendix B.1.2 show that the correlation-based loss outperforms other alternatives. While Flex-KD can in principle be applied across multiple layers, in this work we demonstrate that matching the student's final hidden layer with the teacher's final hidden layer is sufficient to outperform existing baselines.

Formally, the input batch $X$ with $n$ samples is fed simultaneously to $T$ and $S$ to produce the batches of features representation $h^T$ and $h^S$, respectively. Since in the previous subsection, we already identified the indices of the top task-relevant units in the teacher hidden layer, i.e., $E$. From $h^T$, we only select the output of the units of indices from $E$, resulting in a hidden representation $h^{T_{d_S}}$ with the same size as $h^S$. To simplify notations, $h^{T_{d_S}}$ and $h^S$ are assumed to be mean-centered along the batch dimension, such that each unit has mean output $0$ over the batch. Maximizing the cross-correlation along the batch dimension between $h^{T_{d_S}}$ and $h^S$ resulting in minimizing the following loss function:

$$L_{\text{Flex-KD}} = \sum_{m=1}^{d_S} (1 - C_{mm})^2, \tag{5}$$

where $L_{\text{Flex-KD}}$ is the student and teacher features matching loss. $C_{mm}$ is the cross-correlation value between the variables $h^{T_{d_S}}_{i_m}$ and $h^S_m$ and is computed as follows:

$$C_{mm} = \frac{\sum_{j=1}^n h^{T_{d_S}}_{j,i_m} h^S_{j,m}}{\sqrt{\sum_{j=1}^n (h^{T_{d_S}}_{j,i_m})^2} \sqrt{\sum_{j=1}^n (h^S_{j,m})^2}}, \tag{6}$$

where $h^{T_{d_S}}_{.,i_m}$ and $h^S_{.,m}$ are the output feature representations of the unit of index $i_m$ and $m$ from the teacher $T$ and the student $S$, respectively. The final distillation loss of the student model is:

$$L_{KD} = \alpha L_{\text{Flex-KD}} + \beta L_{\text{logit}}, \tag{7}$$

where $L_{\text{logit}}$ is the logit distillation loss, e.g., Gu et al. (2024a); Sanh et al. (2019). $L_{\text{Flex-KD}}$ can be applied as a standalone loss for distillation without $L_{\text{logit}}$. Typically, the final student training loss is:

$$L_{Final} = L_{KD} + \lambda L_1, \tag{8}$$

Table 1: Test accuracy (%) on the IMDB dataset, averaged over three random seeds. Values in green denote gains over the KD baseline, while values in red indicate drops. For GPT2, distillation is from $h^T = 1024$ to $h^S = 768$; for BERT, from $h^T = 768$ to $h^S = 312$.

| Method | 345M → 124M GPT2 | 110M → 14M BERT |
|---|---|---|
| Teacher | 95.47 | 94.06 |
| FT (Devlin et al., 2019) | $94.20 \pm 0.30$ | $89.24 \pm 0.08$ |
| KD (Hinton, 2015) | $94.21 \pm 0.42$ | $89.58 \pm 0.10$ |
| *Projector* (Jiao et al., 2020) | $94.01 \pm 0.12$ (-0.20) | $89.39 \pm 0.05$ (-0.19) |
| *CKA* (Dasgupta & Cohn, 2025) | $94.65 \pm 0.10$ (+0.44) | $90.13 \pm 0.06$ (+0.55) |
| ***Flex-KD*** | $\mathbf{95.09 \pm 0.04}$ (+0.88) | $\mathbf{90.60 \pm 0.04}$ (+1.02) |

Table 2: Results (in %) averaged over three random seeds. The teacher model is GPT2-medium (345M parameters) and the student model is GPT2-small (124M parameters). "AVG" denotes the average performance across all tasks. Values in green indicate performance gains over the KD baseline, while those in red indicate performance drops. Feature distillation is performed from $h^T = 1024$ to $h^S = 768$. For the full table with standard deviations, see Table 8 in Appendix B.1.

| Method | SST-2 | STS-B | MRPC | RTE | MNLI | AVG |
|---|---|---|---|---|---|---|
| Teacher(12 x 1024) | 94.49 | 88.23 | 84.09 | 68.23 | 85.10 | 84.02 |
| FT (Devlin et al., 2019) | 91.32 | 86.58 | 81.68 | **65.95** | 81.78 | 81.46 |
| KD (Hinton, 2015) | 91.63 | 86.56 | 83.35 | 64.98 | 81.12 | 81.52 |
| *Projector* (Jiao et al., 2020) | 90.88 | 86.66 | **83.73** | 64.14 | 81.98 | 81.47 (-0.05) |
| *CKA* (Dasgupta & Cohn, 2025) | 91.32 | 86.93 | 82.40 | 64.62 | **82.52** | 81.55 (+0.03) |
| ***Flex-KD*** | **92.67** | **87.14** | 83.20 | 64.86 | 82.30 | **82.03** (+0.51) |

where $L_1$ is the supervised training loss, e.g., the cross-entropy loss in classification (Sanh et al., 2019) and the language modeling loss in generation (Dasgupta & Cohn, 2025). $\alpha$, $\beta$, and $\lambda$ are hyper-parameters to control the contribution of each term in the final loss. For instance, $\beta = 0$ corresponds to pure feature distillation.

## 4 EXPERIMENTAL RESULTS

We evaluate our approach across three core tasks: text classification, instruction-following, and summarization. Following Dasgupta & Cohn (2025); Jiao et al. (2020); Sanh et al. (2019), in our work, all teacher models are static during the distillation process.

### 4.1 CLASSIFICATION

In this experiment, we evaluate our approach on the Internet Movie Database (IMDB) dataset (Maas et al., 2011) and several tasks from the GLUE benchmark (Wang et al., 2018). Experimental details and baselines are discussed in Appendix B.1.

For IMDB, as shown in Table 1, our proposed method consistently achieves the highest performance across all models. Specifically, it attains an accuracy of 95.09% with the distilled GPT2-small model and 90.60% with the distilled TinyBERT model. This corresponds to improvements of 0.44% and 1.08% over CKA and Projector, respectively, in the GPT2-small setting. Furthermore, it yields up to 1.21% gain with TinyBERT relative to the Projector baseline. Beyond accuracy, our method also exhibits a lower standard deviation, highlighting its stability and consistency. Notably, while Projector frequently degrade student performance, our approach consistently enhances it.

On the GLUE benchmark, as presented in Table 2, Flex-KD consistently outperforms baselines, achieving the highest average score of 82.03% across five tasks. It surpasses both CKA and Projector on four of five tasks, demonstrating its robustness across diverse language understanding settings. For instance, on SST-2, Flex-KD achieves improvements of 1.35% and 1.87% over CKA and Projector, respectively. While standard fine-tuning (FT) outperforms all distillation-based methods on RTE,

among feature distillation approaches (Projector, CKA, and Flex-KD), ours delivers the best result. Projector, on average, degrades student performance, whereas Flex-KD consistently improves it.

**Units selection strategies.** For completeness, in this experiment, we compare three common approaches for estimating unit importance in LLMs: activation-based methods (Muralidharan et al., 2024; Zhang et al., 2023a), standard gradients (Iurada et al., 2025; Guo et al., 2025; Song et al., 2024), and integrated gradients (Dai et al., 2021). As shown in Figure 3a, while all methods yielded similar overall performance, the standard gradient method achieved the highest performance with the smallest standard deviation. These results support our choice of using standard gradients for unit importance estimation.

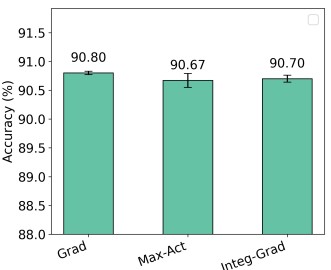 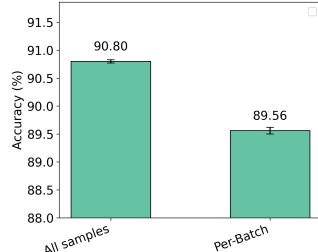 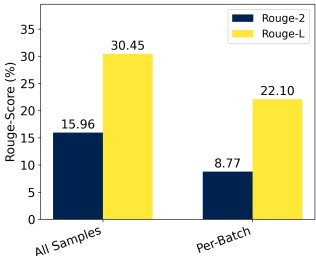

(3a) Flex-KD with different unit se-lection methods on IMDB.

(3b) Flex-KD with two aggregation methods on IMDB classification (**left**) and XSum summarization (**right**).

**Batch vs all samples aggregation.** In our work, during the unit selection stage, we adopt a global aggregation strategy over the entire dataset, as empirical evidence indicates that this approach yields the most efficient training and highest performance. To illustrate this, in Figure 3b, we show that the global aggregation approach consistently outperforms per-batch aggregation by a substantial margin, reaching improvements of up to 8%. We attribute this to the fact that frequently updating the selected nodes at every iteration, or every few iterations, introduces instability during training, which can degrade performance and hinder the student model's ability to focus effectively.

**Sensitivity analysis.** We investigate the impact of the hy-perparameter $\alpha$, which controls the weight of the $L_{\text{Flex-KD}}$ loss, on the student model performance on the IMDB dataset. The final training objective is a weighted combi-nation of $L_{\text{Flex-KD}}$ and the supervised cross-entropy loss, where the weight of the supervised component is fixed at 0.5, and $\alpha$ is varied across the range $[0.05, 0.1, 0.5, 1, 10]$. We used the same setup outlined in Section 4.1. Each experiment is repeated for 3 random seeds and the average is reported. As shown in Figure 4, Flex-KD consistently outperforms the Projector baseline across all $\alpha$ values, demonstrating its robustness and effectiveness.

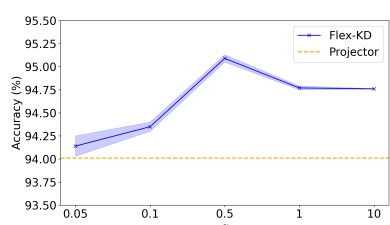

Figure 4: Student model performance on the IMDB dataset as a function of $\alpha$.

## 4.2 INSTRUCTION-FOLLOWING

In this stage, we consider instruction-following (Ouyang et al., 2022) as a conditional text generation task, where models are trained to generate responses conditioned on instructions. A teacher model is fine-tuned on a dataset $D$ comprising instruction–response pairs. We then evaluate various knowledge distillation methods by assessing the instruction-following capabilities of the student model on the same task. Full experimental details and baselines discussions are available in Appendix B.3.

As shown in Table 4, Flex-KD consistently outperforms state-of-the-art baselines across most datasets. For GPT2-small, it achieves the highest average performance, with a 20.45% ROUGE-L score. Notably, it surpasses the Projector baseline on all datasets, with gains of up to 1.23% on S-NI and 0.94% on SelfInst, and outperforms CKA on four of five datasets. Similarly, for OPT-1.3B, Flex-KD delivers the best average performance across datasets, exceeding Projector by up to 2.1% on UnNI and 1.79% on S-NI, and outperforming CKA on four of five datasets, with gains of 1.52% on UnNI, 1.41% on S-NI, and 1.47% on SelfInst. On LlaMA, Flex-KD leads on three of five datasets and ranks second on the rest. It outperforms Projector and CKA by 1.39% and 2.71% on UnNI, and by 0.64% and 2.89% on S-NI, respectively. Overall, while Projector and CKA often degrade OPT and

Table 3: The Rouge-L score (%) of the different approaches. * means the result is reported from Gu et al. (2024a). #Pars represents the number of parameters. "AVG" represents the average performance across all evaluated tasks. Values highlighted in green denote positive performance gains relative to the KD baseline, whereas values in red indicate negative changes.

| Model | #Pars | Method | Dolly | SelfInst | Vicuna | S-NI | UnNI | AVG |
|---|---|---|---|---|---|---|---|---|
| Llama | 7B | Teacher | 28.85 | 20.89 | 18.88 | 32.88 | 36.48 | 27.60 |
| | 1.3B | FT* (Devlin et al., 2019) | 25.85 | 14.59 | 17.41 | 24.13 | 28.22 | 22.04 |
| | | KD* (Hinton, 2015) | **26.17** | 15.13 | 17.34 | 24.97 | 29.22 | 22.57 |
| | | SeqKD* (Kim & Rush, 2016) | 25.98 | 15.00 | 17.66 | 25.36 | 29.83 | 22.76 |
| | | MiniLLM (Gu et al., 2024a) | 25.66 | 15.01 | 18.34 | 27.89 | 32.39 | 23.86 |
| | | *Projector* (Jiao et al., 2020) | **26.17** | 17.15 | **19.12** | 30.59 | 34.19 | 25.44 (+1.58) |
| | | *CKA* (Dasgupta & Cohn, 2025) | 25.63 | 15.83 | 18.20 | 28.34 | 32.87 | 24.17 (+0.31) |
| | | ***Flex-KD*** | 25.92 | **17.21** | 18.91 | 31.23 | **35.58** | **25.77** (+1.91) |
| GPT2 | 1.5B | Teacher | 27.20 | 13.55 | 17.02 | 27.46 | 32.39 | 22.92 |
| | 120M | FT* (Devlin et al., 2019) | 23.30 | 10.00 | 14.70 | 16.30 | 18.50 | 16.56 |
| | | KD* (Hinton, 2015) | 22.80 | 10.08 | 13.40 | 19.70 | 22.00 | 17.59 |
| | | SeqKD* (Kim & Rush, 2016) | 22.70 | 10.10 | 14.30 | 16.40 | 18.80 | 16.46 |
| | | MiniLLM (Gu et al., 2024a) | 24.18 | **12.33** | **17.92** | 22.67 | 24.60 | 20.34 |
| | | *Projector* (Jiao et al., 2020) | 23.60 | 11.36 | 17.61 | 21.78 | 24.63 | 19.79 (-0.55) |
| | | *CKA* (Guo et al., 2025) | 24.07 | 12.15 | 17.83 | 22.74 | 24.35 | 20.22 (-0.12) |
| | | ***Flex-KD*** | **24.45** | 12.30 | 17.62 | **23.01** | **24.87** | **20.45** (+0.11) |
| OPT | 6.7B | Teacher | 28.48 | 16.74 | 18.23 | 29.92 | 32.64 | 25.20 |
| | 1.3B | FT* (Devlin et al., 2019) | **26.00** | 11.40 | 15.60 | 23.10 | 28.40 | 20.90 |
| | | MiniLLM (Gu et al., 2024a) | 25.50 | 13.54 | **17.47** | 24.57 | 27.46 | 21.70 |
| | | *Projector* (Jiao et al., 2020) | 25.44 | 12.77 | 16.84 | 23.98 | 26.62 | 21.13 (-0.57) |
| | | *CKA* (Dasgupta & Cohn, 2025) | 25.25 | 12.84 | 17.25 | 24.36 | 27.20 | 21.38 (-0.32) |
| | | ***Flex-KD*** | 25.54 | **14.31** | 17.06 | **25.77** | **28.72** | **22.28** (+0.58) |

GPT performance, Flex-KD consistently improves it. On LlaMA, although Projector outperforms CKA, Flex-KD still surpasses both, consistently achieving the best overall results.

## 4.3 EVALUATION WITH OTHER LOGIT DISTILLATION LOSSES

In our main instruction-following experiments, all feature-distillation methods (Projector, CKA, and Flex-KD) were combined with the standard reverse KL loss from MiniLLM. To evaluate whether Flex-KD remains effective under alternative logit-based objectives, we further pair each feature-distillation method with two recent and strong baselines: GKD (Agarwal et al., 2024) and DistiLLM (Ko et al., 2024a). As shown in Table 4, Flex-KD consistently provides the largest improvements over both logit KD methods, yielding gains of up to +1.10 Rouge-L with GKD and +0.90 with DistiLLM. It also outperforms the competing feature-alignment approaches, with projector-based distillation significantly degrading performance and CKA offering only marginal gains. These results demonstrate that Flex-KD is complementary to diverse logit-level KD formulations and serves as a stable, architecture-agnostic feature-alignment mechanism that strengthens any underlying logit distillation objective.

## 4.4 SUMMARIZATION

Following Dasgupta & Cohn (2025), in this subsection, we distill large encoder–decoder models on the task of single-document news summarization. Specifically, we distill BART-large (Lewis et al., 2019) into a set of smaller student architectures, varying in depth (6 and 12 layers) and hidden dimensionality (640 and 768), and evaluate on the CNN/DailyMail (Hermann et al., 2015) and XSum (Narayan et al., 2018) datasets. Experimental details and baselines are discussed in Appendix B.4.

As shown in Table 5, student models trained with our Flex-KD approach consistently achieve superior ROUGE scores compared to all baselines. Under a 5.5× compression ratio, Flex-KD demonstrates a substantial performance improvement, achieving up to a 3.75-point increase in ROUGE-L score (RL) over Projector, which notably degrades student performance relative to the logit KD (KD) baseline on the CNN/DailyMail and XSum datasets. Flex-KD also outperforms the CKA baseline on both datasets, yielding improvements of up to 0.58% as ROUGE-2 (R2) and 0.82% as RL on the

Table 4: Rouge-L results on S-NI and UnNI for Projector, CKA, and Flex-KD combined with GKD and DistiLLM. Values highlighted in green denote positive performance gains relative to the logit baseline, whereas values in red indicate negative changes.

| Model | #Pars | Method | S-NI | UnNI | AVG |
|---|---|---|---|---|---|
| | 1.5B | Teacher | 27.46 | 32.39 | 29.92 |
| GPT2 | 120M | GKD (Agarwal et al., 2024) | 18.88 | 21.41 | 20.14 |
| | | *Projector + GKD* (Jiao et al., 2020) | 15.86 | 16.44 | 16.15 (-3.99) |
| | | *CKA + GKD* (Guo et al., 2025) | 19.52 | 21.62 | 20.57 (+0.43) |
| | | ***Flex-KD + GKD*** | **19.91** | **22.58** | **21.24** (+1.10) |
| | | Distillm (Gu et al., 2024a) | 25.04 | 27.68 | 26.36 |
| | | *Projector + Distillm* (Jiao et al., 2020) | 21.60 | 23.74 | 22.67 (-3.69) |
| | | *CKA + Distillm* (Guo et al., 2025) | 25.55 | 27.55 | 26.55 (+0.19) |
| | | ***Flex-KD + Distillm*** | **26.52** | **28.00** | **27.26** (+0.90) |

Table 5: ROUGE-2 (R2) and ROUGE-L (RL) scores for different BART students on the CNN/DailyMail and XSum datasets. Every BART student has an equal number of encoder and decoder layers. All baseline results are taken from Dasgupta & Cohn (2025). Values in green indicate a positive performance gain over the KD baseline. P(M) is number of parameters in Million. C.R. is the compression ratio.

| Model | #P(M) | C.R. | R2(CNN) | RL(CNN) | R2(XSum) | RL(XSum) |
|---|---|---|---|---|---|---|
| BART-large (24 × 1024) | 440 | 1.0× | 21.00 | 30.60 | 21.80 | 36.50 |
| KD (6 × 640) (Hinton, 2015) | 80 | 5.5× | 15.10 | 25.80 | 13.50 | 27.40 |
| *Projector* (6 × 640) (Jiao et al., 2020) | 80 | 5.5× | 14.80 (-0.30) | 25.60 (-0.20) | 12.70 (-0.80) | 26.70 (-0.70) |
| *CKA* (6 × 640) (Dasgupta & Cohn, 2025) | 80 | 5.5× | 16.80 (+1.70) | 26.80 (+1.00) | 15.00 (+1.50) | 29.20 (+1.80) |
| ***Flex-KD*** (6 × 640) | 80 | 5.5× | **17.38** (+2.28) | **27.62** (+1.82) | **15.96** (+2.46) | **30.45** (+3.05) |
| KD (6 × 768) (Hinton, 2015) | 100 | 4.4× | 16.40 | 26.80 | 15.10 | 29.20 |
| *Projector* (6 × 768) (Jiao et al., 2020) | 100 | 4.4× | 15.50 (-0.90) | 26.20 (-0.60) | 14.10 (-1.00) | 28.20 (-1.00) |
| *CKA* (6 × 768) (Dasgupta & Cohn, 2025) | 100 | 4.4× | 17.70 (+1.30) | 27.70 (+0.90) | 16.50 (+1.40) | 31.00 (+1.80) |
| ***Flex-KD*** (6 × 768) | 100 | 4.4× | **17.70** (+1.30) | **27.96** (+1.16) | **16.65** (+1.55) | **31.13** (+1.93) |

CNN/DailyMail dataset. Gains are even more significant on the XSum dataset, with improvements of up to 0.96% (R2) and 1.25% (RL). Under a 4.4× compression ratio, Flex-KD again surpasses both Projector and CKA, delivering RL improvements of up to 2.93% over Projector.

Although Flex-KD transfers knowledge only between the final encoder and decoder layers of the teacher and student, it consistently outperforms multi-layer distillation methods (Projector and CKA) that transfer knowledge to all student layers, maintaining superior performance even under substantial compression ratios. Results with deeper layers are reported in Table 10 of Appendix B.4.

**Multi-Layer distillation.** In Figure 5a, we evaluate Flex-KD under a multi-layer distillation setup on XSum, where knowledge is transferred from multiple teacher layers to student layers. T and S are the total number of teacher and student hidden layers. The x-axis illustrates the element-wise mapping between teacher and student layers. Our results show that applying Flex-KD solely on the final hidden layer is sufficient to achieve strong performance, leading to the best performance/efficiency balance. This contrasts with methods such as CKA (Dasgupta & Cohn, 2025), which require distillation across all hidden layers. We argue that multi-layer distillation introduces a more complex optimization process, whereas last-layer distillation is simpler and more efficient. Prior work has also highlighted the importance of the final hidden state in LLMs, showing its significance for both generative and classification tasks (Gromov et al., 2024; Men et al., 2024; Saadi et al., 2023a).

**Ablation study.** In Figure 5b, we conduct an ablation study evaluating the contribution of each loss component on XSum summarization, using BART (6×640) as the student architecture. The results demonstrate that all three components of the final loss are essential, as their combination yields the strongest overall performance. In particular, our proposed Flex-KD loss ($L_{Flex-KD}$) provides substantial gains, improving ROUGE-2 by 2.46% and ROUGE-L by 3.05%.

**Compute Cost.** Flex-KD introduces only one additional operation beyond standard logit-based KD: a *single* gradient-based pass to compute unit importance from the teacher's final hidden layer. This step is performed once before training and therefore does not scale with the number of epochs or the depth of the teacher model. As reported in Appendix B.4, Table 11, this selection stage

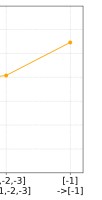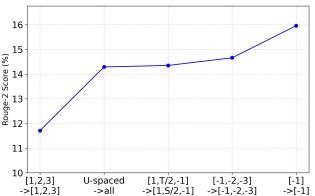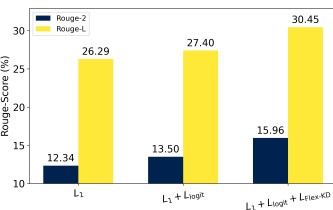

(5a) Flex-KD with different layers distillation.      (5b) Flex-KD loss components.

adds only ∼26 seconds on XSum and ∼23 seconds on CNN/DailyMail. During training, Flex-KD applies feature alignment only on the last encoder/decoder layers, resulting in significantly lower per-epoch cost than full-layer feature distillation methods. Table 12 (Appendix B.4) shows that Flex-KD matches the efficiency of pure logit KD (0.93 hours/epoch) and is markedly faster than both Projector (1.94 h/epoch) and CKA-KD (1.35 h/epoch), demonstrating that Flex-KD provides the most computationally efficient feature-distillation strategy among the baselines.

## 4.5 ROBUSTNESS TO DATA SCARCITY

Flex-KD relies on access to training data in order to compute task-relevant unit importance scores through gradient attribution. A natural concern is whether its performance degrades significantly when only limited data is available for this computation. To investigate that, we conduct an experiment using only a small subset, 5% the Dolly dataset. We follow the same setup of the GPT2 in Subsection 4.2, with the exception that the teacher model is fine-tuned for five epochs, and distillation is performed for 1,000 steps. Despite this significant reduction in supervision signal, Flex-KD is still able to identify meaningful units subsets and consistently outperforms baseline approaches. As shown in Table 6, our method achieves an average Rouge-L score of 19.01%, outperforming both baselines, which obtain average scores of 18.08% and 18.68%, respectively. Notably, Flex-KD also matches or surpasses other methods across most individual tasks, demonstrating its robustness even in low-resource scenarios.

Table 6: The Rouge-L score of the different approaches with 5% of the dolly data.

| Method | Dolly | SelfInst | Vicuna | S-NI | UnNI | AVG |
|---|---|---|---|---|---|---|
| Teacher | 21.84 | 12.74 | 15.63 | 22.87 | 25.99 | 19.81 |
| Projector (Jiao et al., 2020) | 22.13 | 10.72 | 16.95 | 19.80 | 20.81 | 18.08 |
| CKA (Dasgupta & Cohn, 2025) | 22.51 | **10.89** | **17.65** | 20.37 | 21.98 | 18.68 |
| **Flex-KD** | **23.07** | 10.84 | 17.09 | **21.50** | **22.58** | **19.01** |

## 5 CONCLUSION

Feature-level knowledge distillation has long promised richer information transfer from teacher to student for large language model compression, but in practice, it has been constrained by the restrictive assumption of matched teacher–student hidden sizes or by learned projectors that introduce parameters and often distort representations. In this work, we introduced Flex-KD, a parameter-free, task-driven framework that overcomes these limitations by selecting and transferring only the most task-relevant teacher dimensions. In doing so, Flex-KD directs the student's limited capacity toward informative signals from the teacher, enabling effective and flexible feature distillation across heterogeneous model sizes. Extensive experiments across classification, summarization, and instruction-following tasks, spanning 13 different datasets and 8 models, demonstrate **consistent** improvements, even in low-data regimes, showing that feature-level KD, with Flex-KD, is both practical and broadly beneficial for LLM compression. Future work includes extending Flex-KD to vision and exploring its application beyond standard transformer architectures, such as to Mamba or between completely heterogeneous network architectures.

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

## A ADDITIONAL FIGURES

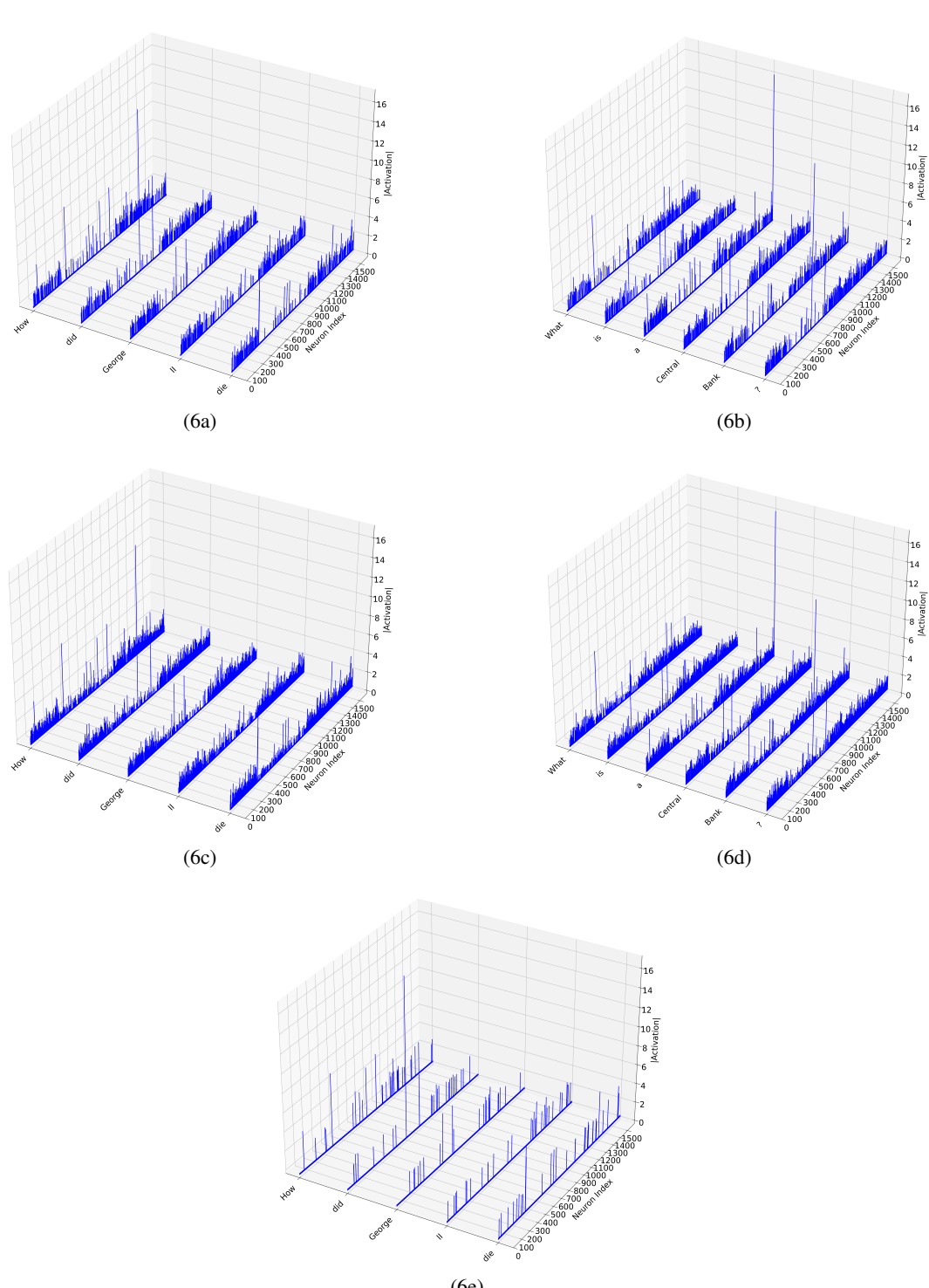

(6a)

(6b)

(6c)

(6d)

(6e)

Figure 6: Activation magnitudes (z-axis) after feeding training samples from the downstream task to a fine-tuned GPT-xlarge. x and y axes are sequence and feature dimensions, respectively: (a) We threshold values below 1 to zero. (b) We threshold values below 0.5 to zero. (c) We threshold values below 2 to zero.

In Figure 6, we visualize the activations of the last hidden layer of a fine-tuned GPT-xlarge model on downstream examples. In 6a and 6b, we threshold values $< 1$ to 0. In 6c and 6d, we threshold values $< 0.5$ to 0. In 6e, we threshold values $< 2$ to 0. As it can be seen, many units display near-zero or low-magnitude activations, suggesting limited contribution to the final output.

## B  Experimental details and additional Results

### B.1  Classification

In this setup, we distill the last layer of the teacher to the last layer of the student. Each method is added as a stand-alone regularizer to the student's standard supervised training loss. This stand-alone performance evaluation allows us to assess the effectiveness of Flex-KD compared to the baselines. For comparison, we also report the performance of the student model trained via standard fine-tuning (FT), without any distillation.

#### B.1.1  Experimental Setup

**On IMDB.** For the IMDB, we use two distinct teacher–student model pairs. In the first setting, we use BERT-base-uncased (110M parameters) (Devlin et al., 2019), fine-tuned on IMDB, as the teacher and TinyBERT-General-4L-312D (14M parameters) (Jiao et al., 2020) as the student. In the second setting, GPT2-medium (345M parameters) (OpenAI, 2023), fine-tuned on IMDB, serves as the teacher, with GPT2-base (124M parameters) (OpenAI, 2023) as the student. The test classification accuracy is reported as the evaluation metric. The teacher models are fine-tuned on the IMDB dataset for 3 epochs with a batch size of 8 and an Adam optimizer with a learning rate equal to $5e-5$. During the distillation process, the student models are trained for 3 epochs, the batch size is set to 8, the optimizer is set to Adam with a learning rate of $5e-5$. Each experiment is repeated for 3 times and the average and the standard deviations are reported. The weight of each KD stand-alone loss and the weight of the hard loss are fixed to 0.5 (Sanh et al., 2019; Jiao et al., 2020), for instance, for Flex-KD, we have $\alpha = 0.5$, $\beta = 0$, and $\lambda = 0.5$.

**On GLUE.** For the GLUE benchmark, we selected 5 datasets that cover different categories and sizes, small-size (RTE, STS-B), medium-size (MRPC, SST-2), and large-size (MNLI), to ensure varied scenarios. MRPC and STS-B for paraphrase and semantic similarity. SST-2 for sentiment classification, and MNLI and RTE for natural language inference. We report results for CKA (Dasgupta & Cohn, 2025), which is a recently proposed feature distillation method that does not require the student and the teacher to have equal hidden dimensions, linear projection (Projector) (Jiao et al., 2020), vanilla KD (KD) (Hinton, 2015), and our proposed approach (Flex-KD). In this setting, we use GPT2-medium (345M parameters), fine-tuned on each of the tasks, as the teacher and GPT2-base (124M parameters) as the student. For MRPC, we report the average of F1 score and accuracy; for STS-B, we report the average of Pearson and Spearman correlations. Accuracy is used as the evaluation metric for the remaining tasks. Here, the teacher is trained for 3 epochs with batch size 8, and and Adam optimizer with a learning rate of $5e-5$. For the distillation process, the number of epochs, the batch size, the learning rate are set to 3, 16, and $5e-5$, respectively. The weight of each KD stand-alone loss and the weight of the hard loss are fixed to 0.5 (Sanh et al., 2019; Jiao et al., 2020), for instance, for Flex-KD, $\alpha = 0.5$, $\beta = 0$, and $\lambda = 0.5$. For the vanilla-KD (KD), the logit loss weight was set to 0.1. All experiments are repeated for three random seeds and the average and the standard deviations are reported. For fair comparison, in this task, we used the same correlation loss as for our proposed method for the projector approach. As demonstrated in prior work (Chen et al., 2022; Jiao et al., 2020), projecting the student's feature representations into the teacher's feature space yields superior performance compared to the reverse direction. Accordingly, in our implementation of the projector-based approach, we align the student's hidden representations to those of the teacher via a learned projection.

#### B.1.2  Loss

In Table 7, we conduct an experiment comparing the performance of three feature distillation loss functions: MSE, cosine distance, and a correlation-based loss on the IMDB classification task. The teacher model is BERT-base (110M) and the student model is TinyBERT (14M). As shown in the

table, the student model trained with the correlation-based loss achieves better performance and exhibits lower standard deviation, demonstrating its effectiveness.

Table 7: Comparison between MSE, Cosine distance and Correlation as feature loss functions.

| Method | 110M $\to$ 14M (BERT) | | |
| | MSE | Cosine | Correlation |
| --- | --- | --- | --- |
| Teacher | | 94.14 | |
| **Flex-KD** | $89.94 \pm 0.07$ | $90.05 \pm 0.13$ | $\mathbf{90.80 \pm 0.03}$ |

### B.1.3   PERFORMANCE ON GLUE

In Table 8, we present the performance of different KD approaches on several tasks from the glue benchmark. Results (in %) are averaged over three random seeds. The teacher model is GPT2-medium (345M parameters), and the student model is GPT2-small (124M parameters). "AVG" represents the average performance across all evaluated tasks. For feature distillation, here we distill from a hidden size of 1024 to a hidden size of 768.

Table 8: Results (%) are averaged over 3 random seeds. M for million. Teacher is gpt2-medium and Student is gpt2-small. AVG is for the average performance across all the tasks.

| Method | SST-2 | STS-B | MRPC | RTE | MNLI | AVG |
| --- | --- | --- | --- | --- | --- | --- |
| Teacher | 94.49 | 88.23 | 84.09 | 68.23 | 85.10 | 84.02 |
| FT | $91.32 \pm 0.29$ | $86.58 \pm 0.33$ | $81.68 \pm 1.34$ | $\mathbf{65.95 \pm 2.45}$ | $81.78 \pm 0.11$ | 81.46 |
| KD | $\underline{91.63 \pm 0.11}$ | $86.56 \pm 0.29$ | $\underline{83.35 \pm 0.87}$ | $\underline{64.98 \pm 0.00}$ | $81.12 \pm 0.06$ | 81.52 |
| Projector | $90.88 \pm 0.72$ | $86.66 \pm 1.45$ | $\mathbf{83.73 \pm 0.54}$ | $64.14 \pm 1.45$ | $81.98 \pm 0.27$ | 81.47 |
| CKA | $91.32 \pm 0.77$ | $\underline{86.93 \pm 0.03}$ | $82.40 \pm 1.43$ | $64.62 \pm 0.00$ | $\mathbf{82.52 \pm 0.27}$ | $\underline{81.55}$ |
| **Flex-KD** | $\mathbf{92.67 \pm 0.13}$ | $\mathbf{87.14 \pm 0.21}$ | $83.20 \pm 1.19$ | $64.86 \pm 1.98$ | $\underline{82.30 \pm 0.21}$ | $\mathbf{82.03}$ |

### B.2   ABLATION ON TEACHER MODELS

### B.3   INSTRUCTION-FOLLOWING

The conducted experiments follow a similar setup to the one outlined in Gu et al. (2024a). We evaluate our method on three teacher–student model pairs. First, the Llama model (7B parameters), fine-tuned on the instruction-following Dolly dataset [1], serves as the teacher, with Llama (1.3B parameters) as the student . In the second setting, the GPT2-XL model (1.5B parameters), fine-tuned on the instruction-following Dolly dataset, serves as the teacher, with GPT2-small (124M parameters) as the student. In the third setting, the OPT-6.7B model, fine-tuned on the Dolly dataset, is distilled into the smaller OPT-1.3B student model. We compare our approach against several approaches, including the standard fine-tuning (FT) of the student model, KD (Sanh et al., 2019), SeqKD (Taori et al., 2023), MiniLLM (Gu et al., 2024a), as well as the direct competitive feature KD methods, i.e., Projector (Jiao et al., 2020) and CKA (Dasgupta & Cohn, 2025). For evaluation metrics, similar to Gu et al. (2024a), we report the Rouge-L (Lin, 2004) score on the following benchmark datasets: Dolly test set, SelfInst (Wang et al., 2022a), Vicuna (Chiang et al., 2023), S-NI (Wang et al., 2022b), and UnNI (Honovich et al., 2023) datasets. The Rouge-L score measures the precision of the model generation and it was shown by Wang et al. (2022b) that it is suitable for large-scale instruction-following evaluation.

Across all settings, we adopt a consistent distillation framework. The student model is first fine-tuned for 3 epochs, and the checkpoint with the lowest validation loss is used as the initialization point for subsequent distillation. The distillation process is run for 5,000 iterations with a total batch size of 8, using the Adam optimizer configured with an $\epsilon = $ 1e-8, and a weight decay of 1e-6. The learning rate is set to 5e-6. All reported results are averaged over three random seeds (10, 20, 30) for training and five seeds (10, 20, 30, 40, 50) for evaluation. Except for Llama we only did the evaluation

---

[1]https://github.com/databrickslabs/dolly/tree/master

Table 9: Unified ablation study across RTE, STS-B, and SST-2. For STS-B, we report the average of Pearson and Spearman. All $\Delta$ values are rounded to three decimal places.

| Ablation Type | % Removed | RTE | | STS-B (Avg) | | SST-2 | |
|---|---|---|---|---|---|---|---|
| | | Acc | $\Delta$ Acc | Avg | $\Delta$ Avg | Acc | $\Delta$ Acc |
| High Importance | 0% | 68.23 | - | 88.23 | - | 94.49 | - |
| | 10% | 60.29 | -7.9 | 87.09 | -1.1 | 94.27 | -0.2 |
| | 20% | 55.23 | -13.0 | 87.09 | -1.1 | 94.15 | -0.3 |
| | 50% | 54.51 | **-13.7** | 81.75 | **-6.5** | 93.46 | **-1.0** |
| Low Importance | 0% | 68.23 | - | 88.23 | - | 94.49 | - |
| | 10% | 68.59 | +0.4 | 88.23 | 0.0 | 94.50 | 0.0 |
| | 20% | 68.23 | 0.0 | 88.21 | 0.0 | 94.50 | 0.0 |
| | 50% | 68.59 | +0.4 | 88.23 | 0.0 | 94.15 | -0.3 |
| Random | 0% | 68.23 | - | 88.23 | - | 94.49 | - |
| | 10% | 68.95 | +0.7 | 88.22 | 0.0 | 94.38 | -0.1 |
| | 20% | 69.68 | +1.4 | 88.12 | -0.1 | 93.92 | -0.6 |
| | 50% | 66.43 | -1.8 | 87.96 | -0.3 | 94.27 | -0.2 |

across 5 seeds, i.e.,(10, 20, 30, 40, 50). The final evaluation is always conducted using the last saved checkpoint.

For the MiniLLM baseline, we employ only the reverse Kullback-Leibler divergence distillation loss, as outlined in Gu et al. (2024a), between the teacher and student logits as the training objective for the student model. For our method (Flex-KD), as well as the CKA and Projector baselines, we use a combination of the logit distillation loss used to train MiniLLM (RKLD) and the corresponding feature-level distillation loss for the student training. For Feature distillation methods, we distill only the teacher last hidden layer to the student last hidden layer.

Specifically, for Flex-KD, $\alpha = 0.05$. Following the configurations from Dasgupta & Cohn (2025), both CKA and projector baselines use a feature loss weighted by 1. The logit-level loss (reverse KL divergence) is consistently weighted by 1 across all methods. In the projector setup, we used mean squared error (MSE) as the loss function, in line with Dasgupta & Cohn (2025); Jiao et al. (2020). For teacher model fine-tuning, all teachers are trained for 10 epochs. GPT2 uses a batch size of 8 and a learning rate of 1e-4, while OPT and Llama are trained with a total batch size of 8 and a learning rate of 1e-5.

The following are some details related to the competitive methods:

- **FT** (Devlin et al., 2019) refers to standard fine-tuning.
- **KD** (Sanh et al., 2019) namely, word-level KD, where the student model is trained on the teacher model's output at each token step.
- **SeqKD** (Taori et al., 2023) refers to sequence-level knowledge distillation, where the student model is trained on data generated by the teacher model.
- **MiniLLM** (Gu et al., 2024a) employs reverse KL divergence to distill knowledge from the teacher model's logits.

We evaluate our models on the following instruction-following datasets:

- **Dolly**: 500 samples from the `databricks-dolly-15K` dataset used as test set.
- **SelfInst** (Wang et al., 2022a): A user-oriented instruction-following set consisting of 252 samples.
- **Vicuna** (Chiang et al., 2023): The set of 80 difficult questions used for the Vicuna evaluation.
- **S-NI** (Wang et al., 2022b): The SUPER-NATURALINSTRUCTIONS test set comprises 9K samples spanning 119 tasks. Following Gu et al. (2024a), we divide it into three subsets based on ground truth response lengths: $[0, 5], [6, 10], [11, +\infty]$ and we use the $[11, +\infty]$ subset.

- **UnNI** (Honovich et al., 2023): The core set of UNNATURALINSTRUCTIONS comprises 60K samples. Following a similar approach to S-NI, we evaluate on a randomly selected subset of 10K examples from the $[11, +\infty]$ range.

### B.4 SUMMARIZATION

In this experiment, we follow a similar experimental setup to that outlined in Dasgupta & Cohn (2025). The student training objective consists of three components: (1) a supervised cross-entropy loss on the target summary, (2) a logit distillation loss, which is the Kullback-Leibler (KL) divergence loss between the teacher and student output distributions, and (3) a feature distillation loss. For the feature distillation, we evaluate three methods: our proposed Flex-KD, CKA (Dasgupta & Cohn, 2025), and a linear projection-based mean squared error (MSE) (Projector) (Dasgupta & Cohn, 2025; Jiao et al., 2020) . Additionally, we report results for standard logit-level KD (Sanh et al., 2019) without any feature distillation.

As described in Dasgupta & Cohn (2025), CKA and Projector losses are applied between each student layer and uniformly spaced layers from the teacher model. For the Projector variant, hidden states from the student and teacher are aligned via learned linear projections, followed by MSE as in Jiao et al. (2020); Dasgupta & Cohn (2025). For our Flex-KD, $L_{\text{Flex-KD}}$ equation 5 is only applied between the last encoder layers and the last decoder layers of the student and the teacher models.

For Flex-KD, we utilize only 640 samples (40 batches) to identify the top task-relevant units in the teacher model and the hyperparameters are set as follows: $\alpha = 0.05$, $\beta = 1$, $\lambda = 1$, and the batch size is 16. Following the setup in Dasgupta & Cohn (2025), training is performed using the Adam optimizer with a learning rate of $1 \times 10^{-4}$ and a weight decay of $5 \times 10^{-4}$. The maximum input context length is set to 1,024 tokens, and the output summary is constrained to 128 tokens. All experiments are conducted on a single NVIDIA A100 GPU with 80 GB of memory.

#### B.4.1 SUMMARIZATION ON XSUM: STUDENT MODEL WITH DEEPER LAYERS

To further evaluate Flex-KD robustness with deeper student models, Table 10 reports results on the XSum dataset, where Flex-KD achieves consistent improvements over the baselines, including up to a 1.43-point gain in ROUGE-L over Projector.

Table 10: R2 and RL for deeper BART students on the XSum dataset. All baseline results are taken from Dasgupta & Cohn (2025). Teacher is BART-Large ($24 \times 1024$) and student is BART ($12 \times 768$).

| Model | R2(XSum) | RL(XSum) |
|---|---|---|
| BART-large | 21.80 | 36.50 |
| KD (Hinton, 2015) | 17.60 | 32.00 |
| *Projector* (Jiao et al., 2020) | 17.70 (+0.10) | 32.10 (+0.10) |
| *CKA* (Dasgupta & Cohn, 2025) | 18.70 (+1.10) | 33.50 (+1.50) |
| ***Flex-KD*** | **18.93** (+1.33) | **33.53** (+1.53) |

#### B.4.2 SUMMARIZATION ON XSUM AND CNN/DAILYMAIL: OVERLAP OF SELECTED UNITS ACROSS RANDOM SEEDS

In Figure 7, we evaluate the stability and consistency of the selected units across five random seeds. As described in the experimental setup, for the CNN/DailyMail and XSum datasets, we randomly sampled 40 batches of examples to compute the task-relevant units. Thus, in this case, checking the consistency of unit selection remains important. As shown in the following figures, the 5 lists of indices obtained from the five random seeds exhibit a high overlap, with 91.5% for XSum and 96.3% for CNN/DailyMail, demonstrating that our unit selection strategy is both stable and consistent, even with a limited number of samples.

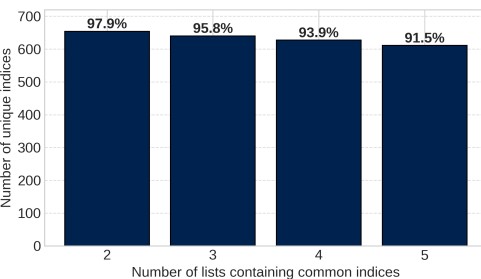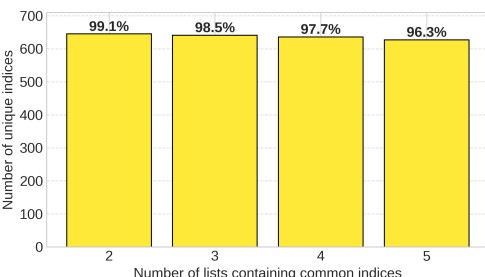

Figure 7: Overlap of selected units across 5 random seeds on (Left) XSum dataset and (right) CNN/DailyMail dateset.

### B.4.3 SUMMARIZATION ON XSUM AND CNN/DAILYMAIL: COMPUTE COST

It is important to note that Flex-KD involves a one-time computation to identify task-relevant units from the teacher model. Since this step is performed only once, it does not pose efficiency concerns. Nevertheless, in Table 11, we conduct an experiment to assess the compute overhead of Flex-KD on the summarization tasks, which involves selecting the task-relevant units for distillation. In this setting, the teacher model is BART-large with 406M parameters, and all experiments were run on a single A100 GPU. In our proposed method, units are selected only from the teacher's last hidden layer. The results show that the additional gradient-based selection step introduces a negligible increase in runtime, around 26 seconds for XSum and 23 seconds for CNN/DailyMail, demonstrating that Flex-KD remains efficient in practice.

Table 11: Compute overhead, results are averaged over 5 random seeds.

|  | **XSum** | **CNN/DailyMail** |
| --- | --- | --- |
| time (seconds) | $25.98 \pm 1.27$ | $22.80 \pm 2.81$ |

Table 12: Computational cost (hours per epoch) of different feature distillation methods. All experiments use a $6 \times 640$ BART student and are run on a single A100 GPU. Flex-KD applies feature alignment only on the last encoder and decoder layers (plus logit loss), while CKA and Projector perform full-layer alignment. On XSUM dataset.

| **Method** | **Hours / Epoch** |
| --- | --- |
| KD Hinton (2015) | 0.93 |
| Projector Jiao et al. (2020) | 1.94 |
| CKA-KD Dasgupta & Cohn (2025) | 1.35 |
| **Flex-KD (ours)** | **0.93** |

## C   LLMS USAGE IN THE PAPER

LLMs were used only occasionally to help polish the writing (propose new words, grammar and spelling correction). All technical ideas, experimental designs, analyses, conclusions, writing were developed and carried out entirely by the authors. The authors have full responsibility for the final text.

