# OpenReview forum: "Flexible Feature Distillation for Large Language Models"
_ICLR.cc/2026/Conference — Submitted to ICLR 2026_

### Official Review · Reviewer_GbPn · 2025-10-23

**Soundness:** 3
**Presentation:** 3
**Contribution:** 2
**Rating:** 6
**Confidence:** 4

**Summary:**

This paper proposes **Flex-KD**, a new method for distilling LLMs based on features instead of logits. Traditional KD methods mostly rely on output logits, but this paper argues that **feature-level distillation** is often more effective. Flex-KD is a **parameter-free** approach that dynamically selects and distills important subspaces of the hidden representations, guided by gradient-based importance metrics. It avoids costly projection modules and maintains flexibility across layers and token positions. Experiments on various teacher–student setups show that Flex-KD achieves superior performance compared to logit-based and feature-based KD baselines like MiniLLM, Projector, and CKA.

**Strengths:**

- Good problem statement; the paper clearly identifies a compelling limitation of traditional logit-based KD, its lack of generalizability to feature representations. The motivation to shift toward a more flexible, representation-level distillation is both well-articulated and timely. Flex-KD framework is a simple yet effective approach that provides a parameter-free alternative to other feature-based KD methods.
- The idea of identifying the most important feature dimensions and distilling that subspace is novel and effective to circumvent the representation projection approach. It seems computationally efficient and architecture-agnostic.

**Weaknesses:**

- **Lack of comparisons with recent logit-based KD baselines**: While the paper compares against Projector, CKA, and MiniLLM, it misses some more recent and state-of-the-art distillation methods such as GKD (Agarwal et al., 2024), DistiLLM (Ko et al., 2024), DistiLLM-2 (Ko et al., 2025), and Speculative KD (Xu et al., 2024). These should be comprehensively included in the comparison, since Flex-KD is meant to compete with these in practical settings. Also, the authors should discuss more about why feature-based KD methods are better than logit-based KD methods.
- The authors mention that *Flex-KD can be combined with logit-based KD*, but **no experiments are provided to support this hybrid strategy**. It must be demonstrated how well this integration performs, and how the state-of-the-art KD methods could be combined with Flex-KD framework.
- All experiments are conducted with **same-family teacher–student pairs** (e.g., LLaMA-7B → LLaMA-1.3B, GPT2-1.5B → GPT2-120M). It remains unclear whether cross-family distillation is feasible under Flex-KD, even when the teacher and student have different embedding subspaces. For example, does it still work when distilling from a LLaMA teacher to a GPT2 student?

**Questions:**

See weaknesses

---

> ### Author Response · Authors · 2025-11-28
> **Response to Reviewer GbPn**
>
> We sincerely appreciate your review and your constructive feedback. Blow, we address your concerns point by point.
>
> >**Q1&Q2:** Lack of comparisons with recent logit-based KD baselines: While the paper compares against Projector, CKA, and MiniLLM, it misses some more recent and state-of-the-art distillation methods such as GKD (Agarwal et al., 2024), DistiLLM (Ko et al., 2024), DistiLLM-2 (Ko et al., 2025), and Speculative KD (Xu et al., 2024). These should be comprehensively included in the comparison, since Flex-KD is meant to compete with these in practical settings. Also, the authors should discuss more about why feature-based KD methods are better than logit-based KD methods. The authors mention that Flex-KD can be combined with logit-based KD, but no experiments are provided to support this hybrid strategy. It must be demonstrated how well this integration performs, and how the state-of-the-art KD methods could be combined with Flex-KD framework.
>
> We first emphasize that **feature-based distillation has always been used together with logit-based distillation** since the earliest works (e.g., Projector (Jiao et al. (2020)) patient-KD (Sun et al.(2019)), CKA-KD (Dasgupta & Cohn (2025)).). Our method follows the same established design principle. For fair comparison, in all the experiments, Flex-KD keeps the original logit loss of each baseline intact, using reverse KL-divergence (MiniLLM) for instruction following and forward KL (CKA-KD paper style) for summarization, and adds our proposed hidden-state alignment/the competing feature distillation losses on top.
>
> **To address the reviewer's request for additional logit-based baselines and to show that our approach can be integrated with any logit distillation method to boost student performance**, we added experiments on the two largest instruction-following datasets (S-NI and UnNI) where we combine Flex-KD and the two competing feature distillation baselines (CKA-KD and projector) with **GKD (Agarwal et al., 2024)** and **Distillm (Ko et al., 2024)**. As shown in the following Table: (i) Flex-KD yields **consistent gains** on top of any logit KD baseline (e.g., +1.10 Rouge-L over GKD and +0.90 over Distillm); (ii) Flex-KD **outperforms both Projector and CKA-KD** in every setting; and (iii) Projector significantly **degrades** student performance, whereas Flex-KD consistently **improves** them. These results demonstrate that Flex-KD provides a **more stable and broadly applicable feature-alignment mechanism**, fully complementary to state-of-the-art logit-based KD approaches.
>
> ### Performance on S-NI and UnNI (GPT2 Teacher → GPT2-120M Student)
>
> | Model | #Params | Method                     | S-NI  | UnNI  | AVG   |
> |-------|---------|----------------------------|-------|-------|-------|
> | GPT2  | 1.5B    | Teacher                    | 27.46 | 32.39 | 29.92 |
> |       |         |                            |       |       |       |
> |       | 120M    | GKD                        | 18.88 | 21.41 | 20.14 |
> |       |         | Projector + GKD            | 15.86 | 16.44 | 16.15 (-3.99) |
> |       |         | CKA + GKD                  | 19.52 | 21.62 | 20.57 (+0.43) |
> |       |         | **Flex-KD + GKD**          | **19.91** | **22.58** | **21.24** (+1.10) |
> |       |         |                            |       |       |       |
> |       | 120M    | Distillm                   | 25.04 | 27.68 | 26.36 |
> |       |         | Projector + Distillm       | 21.60 | 23.74 | 22.67 (-3.69) |
> |       |         | CKA + Distillm             | 25.55 | 27.55 | 26.55 (+0.19) |
> |       |         | **Flex-KD + Distillm**     | **26.52** | **28.00** | **27.26** (+0.90) |
>
>
> >**Q3:** All experiments are conducted with same-family teacher–student pairs (e.g., LLaMA-7B →
> LLaMA-1.3B, GPT2-1.5B → GPT2-120M). It remains unclear whether cross-family distillation is
> feasible under Flex-KD, even when the teacher and student have different embedding subspaces. For
> example, does it still work when distilling from a LLaMA teacher to a GPT2 student?
>
> Thank you for raising this point. In this work, we deliberately focused on providing a thorough and controlled evaluation of Flex-KD against competing methods across diverse settings: classification, summarization, instruction following, and low-data regimes, using multiple backbone architectures to isolate and assess the core contribution of task-driven with different hidden size feature distillation. While cross-family distillation is indeed an interesting extension, but its primary challenge lies in tokenizer mismatch, not in any limitation of Flex-KD. Teacher and student models from different families (e.g., LLaMA → GPT2) typically rely on distinct vocabularies and tokenization rules, which prevents straightforward alignment of inputs, logits, or hidden representations. This difficulty affects all KD approaches, including standard logit-based KD and resolving it would require major changes in the setup.

---

### Official Review · Reviewer_uBcB · 2025-10-26

**Soundness:** 2
**Presentation:** 2
**Contribution:** 1
**Rating:** 0
**Confidence:** 4

**Summary:**

This paper introduces Flex-KD, a framework for feature-based knowledge distillation designed to compress LLMs into smaller student models with different hidden state dimensions.

**Strengths:**

The idea of selecting a feature subspace based on gradient-based importance is a clever, parameter-free way to direct the student's limited capacity towards what matters most for a specific task.

**Weaknesses:**

1. The primary weakness lies in the potentially simplistic definition of "task-relevance" and the static nature of the unit selection process. The method relies on a first-order gradient magnitude, which captures local sensitivity but may fail to account for more complex, non-linear, or cooperative interactions between hidden units.
2. The paper only attempts to distill a smaller language model from several very old, small-parameter models. Small models under the latest training paradigms have become more powerful, and the effectiveness of the proposed method under these new models is questionable. Furthermore, the effectiveness of the proposed Flex-KD method using a very large-parameter model as the teacher is questionable.
3. The methods compared in this paper are too old. In the past year, a lot of work has considered feature alignment rather than just logits alignment [1]. Also, because the tasks in this paper on are too simple, the advantage over such simple baselines is very minor.

[1] DDK: Distilling Domain Knowledge for Efficient Large Language Models

**Questions:**

I would suggest the author to conduct experiments based on the most cutting-edge LLM in the next submission.

---

> ### Author Response · Authors · 2025-11-28
> **Response to Reviewer uBcB (1/2)**
>
> >**Q1:** The primary weakness lies in the potentially simplistic definition of "task-relevance" and the static nature of the unit selection process. The method relies on a first-order gradient magnitude, which captures local sensitivity but may fail to account for more complex, non-linear, or cooperative interactions between hidden units.
>
> We would like to clarify that the definition of task-based knowledge distillation(Sun et al., 2019;
> Saadi et al., 2023; Dasgupta & Cohn, 2025) and task-relevant LLM components (Ferrando et al. (2024); Muralidharan et al. (2024); Iurada et al., 2025) follow a well-established line of work in the literature. Task relevance of a unit is typically defined via its influence on the task loss or output (Sundararajan et al. (2017); Nielsen et al. (2022); Ferrando et al. (2024); Muralidharan et al. (2024)). Many recent works identify important units/heads in an LLM using gradient-based sensitivity (Muralidharan et al. (2024); Iurada et al. (2025); Guo et al. (2025). Flex-KD selects the teacher subspace to be distilled by aggregating the gradient magnitude of the output with respect to the final hidden state. **Importantly, We also do not rely on this choice blindly**: we explicitly compared in **Figure 3a of subsection 4.1** alternative attribution strategies. We evaluated (i) gradient magnitudes, (ii) activation magnitudes, and (iii) Integrated Gradients. We found that standard gradients consistently yielded the best downstream performance. Moreover, in a separate experiment, **in Figure 3b of subsection 4.1** of the same section, we compared dynamic selection (recomputing important units at every iteration across small batches) with our static selection (computed once over all the samples), and observed that static selection is not only more efficient but also outperforms dynamic selection by a large margin (**up to 8\%**), suggesting that frequent/batches-based reselection mostly makes the training instable. These results support our design choice.
>
> **Finally, we would like to reposition this discussion within the main goal of our work, as we believe the reviewer may have misunderstood the core contribution.** In the LLM KD literature, most methods focus on logit distillation because classical feature KD requires teacher and student to have the same hidden size, which severely restricts student architecture flexibility. Only tow methods in the literature have tried to solve this limitation: (i) linear projectors (Sun et al., 2019) to match dimensions, which distort teacher embeddings, add parameters, and are known (and confirmed in our experiments) to degrade performance in low-data regimes and generative tasks, and (ii) CKA-based alignment (Dasgupta & Cohn, 2025), which ignores task relevance and aligns all teacher units indiscriminately, leading to limited improvements in several tasks. **Flex-KD is explicitly designed to overcome these limitations.** It introduces a task-driven, projection-free, cross-width feature alignment mechanism that selects only the most relevant teacher units. As shown across classification, instruction-following, and summarization, Flex-KD consistently outperforms both projector-based and CKA-based approaches, demonstrating that task-aware selective alignment provides a substantially more effective solution to the width-mismatch problem than existing alternatives.
>
>
> >**Q2:** The paper only attempts to distill a smaller language model from several very old, small-parameter models. Small models under the latest training paradigms have become more powerful, and the effectiveness of the proposed method under these new models is questionable. Furthermore, the effectiveness of the proposed Flex-KD method using a very large-parameter model as the teacher is questionable.
>
> We respectfully disagree with the reviewer’s point. Our experimental setup follows the standard practice used in recent state-of-the-art KD papers, including CKA-KD  (Dasgupta & Cohn (2025)), ABKD (Wang et al. (2025)), and DistiLLM (Ko et al. (2024)), all of which evaluate feature or logit distillation on teacher–student pairs of \textbf{comparable (or smaller) }scale, from the following models BERT, GPT, OPT, LLaMA, and BART. These model families remain the dominant benchmarks for controlled and reproducible KD research, and are used in top-tier venues (ICLR/ICML) to evaluate new distillation algorithms.

---

> ### Author Response · Authors · 2025-11-28
>
> >**Q3:** The methods compared in this paper are too old. In the past year, a lot of work has considered feature alignment rather than just logits alignment [1]. Also, because the tasks in this paper on are too simple, the advantage over such simple baselines is very minor.
> [1]DDK: Distilling Domain Knowledge for Efficient Large Language Models
>
> We respectfully disagree with the reviewer’s assessment. To the best of our knowledge, the only state-of-the-art feature distillation methods that explicitly address the hidden-dimension mismatch problem in LLMs are Projector (Jiao et al. (2020)) and CKA-KD (Dasgupta & Cohn (2025)). These methods are therefore the directly relevant competitive baselines for Flex-KD. Notably, the most recent of these works  (Dasgupta & Cohn (2025)) (ICLR 2025) also considers Projector as the sole baseline, confirming that these are the appropriate SOTA comparisons for cross-width feature alignment. Beyond feature distillation, we also compare against strong and widely used logit-based KD baselines, including KD, SeqKD, and MiniLLM. To further strengthen the evaluation, we added experiments on the largest instruction-following datasets (S-NI and UnNI) using the latest logit-based KD methods, including DistiLLM (Ko et al., 2024) and GKD (Agarwal et al., 2024) in the following Table
>
> ### Performance on S-NI and UnNI (GPT2 Teacher → GPT2-120M Student) with GKD and Distillllm as logit loss.
>
> | Model | #Params | Method                     | S-NI  | UnNI  | AVG   |
> |-------|---------|----------------------------|-------|-------|-------|
> | GPT2  | 1.5B    | Teacher                    | 27.46 | 32.39 | 29.92 |
> |       |         |                            |       |       |       |
> |       | 120M    | GKD                        | 18.88 | 21.41 | 20.14 |
> |       |         | Projector(Based on GKD)            | 15.86 | 16.44 | 16.15 (-3.99) |
> |       |         | CKA(Based on GKD)                | 19.52 | 21.62 | 20.57 (+0.43) |
> |       |         | **Flex-KD(Based on GKD)**          | **19.91** | **22.58** | **21.24** (+1.10) |
> |       |         |                            |       |       |       |
> |       | 120M    | Distillm                   | 25.04 | 27.68 | 26.36 |
> |       |         | Projector(Based on Distillm)      | 21.60 | 23.74 | 22.67 (-3.69) |
> |       |         | CKA(Based on Distillm)              | 25.55 | 27.55 | 26.55 (+0.19) |
> |       |         | **Flex-KD(Based on Distillm) **     | **26.52** | **28.00** | **27.26** (+0.90) |
>
> Flex-KD consistently provides gains on top of all these methods. **Regarding the reviewer’s suggested paper (DDK: Distilling Domain Knowledge for Efficient LLMs), this paper focuses on completely different problem and setup ( domain knowledge transfer). it does not perform feature alignment or address hidden-dimension mismatch, making it unrelated to the problem Flex-KD is designed to solve and hence it is not a valid baseline.** If the reviewer is aware of additional modern feature-distillation methods that specifically target cross-width hidden-state alignment, we would welcome the pointers.

---

### Official Review · Reviewer_X9ZQ · 2025-10-27

**Soundness:** 3
**Presentation:** 3
**Contribution:** 2
**Rating:** 4
**Confidence:** 5

**Summary:**

This paper introduces Flex-KD, a parameter-free, task-driven feature distillation method for compressing LLMs. Unlike traditional feature-level KD techniques that require matching hidden dimension or rely on learnable, potentially distorting projectors, Flex-KD selects and transfers only the most task-relevant teacher features to the student. It achieves this by computing gradient-based importance scores to identify key dimensions in the teacher's hidden states and distills only the top-scoring subset, enabling effective transfer even when teacher and student have mismatched hidden sizes. Flex-KD integrates seamlessly into existing KD pipelines. Experiments across classification, instruction-following, and summarization on multiple models and datasets demonstrate consistent improvements over prior state-of-the-art baselines, particularly in low-data regimes.

**Strengths:**

1. This paper tackles a well-motivated and underexplored limitation of feature-based KD for LLMs, namely, the rigid requirement for teacher and student hidden size alignment or the pitfalls of linear projection. This directly addresses real-world deployment needs for LLM compression.

2. Flex-KD's selective, gradient-based feature matching is a creative and simple idea, which avoids introducing new parameters and sidesteps projector-induced feature distortion. It encourages the student to focus on the most informative subspace rather than all features, which is both intuitive and empirically validated.

**Weaknesses:**

1. While Flex-KD's selective feature distillation is effective, the manuscript's novelty over CKA and activation/importance-based selection is somewhat incremental. CKA already enables flexible hidden size matching, and the added selectivity via gradient-based ranking, while empirically justified, is a modest step forward.

2. Although Table 9 reports computational overheads, the paper does not compare against logit-based KD, which is the most lightweight baselines. Without this reference, the actual extra cost of Flex-KD's feature distillation / gradient-based selection remains unclear.

3. Flex-KD always selects the top-$d_S$ features globally, which may be suboptimal for tasks with input-dependent importance. The approach does not explore dynamic or per-example selection strategies or consider distributing features in a more nuanced way. This weakens the case that such a hard selection will always work.

**Questions:**

See weaknesses.

---

> ### Author Response · Authors · 2025-11-28
> **Response to Reviewer X9ZQ**
>
> We sincerely appreciate your review and your feedback. Blow, we address your concerns point by point.
>
> >**Q1:** While Flex-KD's selective feature distillation is effective, the manuscript's novelty over CKA and activation/importance-based selection is somewhat incremental. CKA already enables flexible hidden size matching, and the added selectivity via gradient-based ranking, while empirically justified, is a modest step forward.
>
> While Projector, CKA-KD, and our method all aim to solve the same hidden-size constraint in current feature-based KD, the former two inherit structural limitations. Projector layers distort the teacher’s representation, add trainable parameters, and, as confirmed in our experiments, often degrade student performance, especially in generative tasks. CKA-KD does also support cross-width alignment, but it uniformly matches all teacher dimensions, implicitly assuming equal importance. When the student is narrower, this results in transferring substantial task-irrelevant information.
>
> **Flex-KD introduces a fundamentally different mechanism**: a task-driven, parameter-free selection of only the most influential teacher units. This shifts cross-width feature distillation from indiscriminate alignment to selective, task-relevant alignment without projection layers. Empirically, this distinction is meaningful, **Flex-KD consistently outperforms CKA-KD across classification, instruction-following, summarization, and in low-data regime**, demonstrating that principled selectivity yields a non-incremental and practically significant improvement.
>
> >**Q2:** Although Table 9 reports computational overheads, the paper does not compare against logit-based KD, which is the most lightweight baselines. Without this reference, the actual extra cost of Flex-KD's feature distillation / gradient-based selection remains unclear.
>
> Feature-based distillation is always used jointly with logit-based KD in prior work (Jiao
> et al. (2020); Sun et al. (2019), CKA Dasgupta & Cohn (2025)), and Flex-KD follows this standard design by adding a feature-alignment term on top of the original logit loss. The only additional cost unique to Flex-KD is the one-time gradient-based unit selection, which requires a single backward pass up to the final hidden layer (not through all teacher layers) and takes only a few seconds, as reported in Table 9 (appendix).
> During training, Flex-KD does not introduce any expensive operations beyond computing a feature-based loss on the selected subspace. This overhead is negligible compared to the forward/backward passes already required for logit KD. **To provide a direct comparison with both feature-based and pure logit-based KD, we measured the total training time (hours per epoch)** for KD (logit distillation only), and Projector, CKA, or Flex-KD. **As shown below, Flex-KD is the most efficient feature distillation method**, achieving significantly lower cost than both Projector and CKA because it operates only on the last encoder/decoder layers rather than performing full-layer alignment. These results demonstrate that Flex-KD adds only negligeable overhead beyond logit KD and is significantly more efficient than existing feature-alignment baselines.
>
> **Table 1**: Computational cost (hours per epoch) of different feature distillation methods (6×640 BART student, single A100 GPU, XSum dataset).
>
> | **Method**                                | **Hours / Epoch** |
> |-------------------------------------------|--------------------|
> | KD (Hinton et al., 2015)                  | 0.93               |
> | Projector (Jiao et al., 2020)             | 1.94               |
> | CKA-KD (Dasgupta et al., 2025)            | 1.35               |
> | **Flex-KD (ours)**                        | **0.93**           |
>
> >**Q3:** Flex-KD always selects the top features globally, which may be suboptimal for tasks with input-dependent importance. The approach does not explore dynamic or per-example selection strategies or consider distributing features in a more nuanced way. This weakens the case that such a hard selection will always work.
>
> Thank you for raising this point. We agree that input-dependent importance is, in principle, an interesting direction. However, as shown in Figure 3b, we already evaluated dynamic (batch-wise) selection versus our global, dataset-level aggregation strategy. Empirically, global selection yields both higher performance (up to 8\% improvement) and more stable training. Our intuition is that frequently changing the selected neurons at every (or every few) iteration introduces instability during training, which degrades performance and prevents the student from effectively focusing.
>
>
> We hope this clarification addresses your concerns, and we kindly invite the reviewer to reconsider the evaluation in light of these additional clarification and results.

---

### Official Review · Reviewer_HhK4 · 2025-11-02

**Soundness:** 3
**Presentation:** 3
**Contribution:** 3
**Rating:** 2
**Confidence:** 4

**Summary:**

This paper introduces Flex-KD, a parameter-free, task-driven feature distillation framework for large language models (LLMs).Unlike conventional feature-based knowledge distillation (KD) methods that assume identical hidden sizes between teacher and student models—or rely on learnable linear projectors to bridge them—Flex-KD identifies the most task-relevant subspace of the teacher’s hidden representations using gradient-based importance scores, and distills only that subset into the student model.The method allows flexible hidden-size matching and avoids projector-induced distortion or extra parameters. The authors integrate Flex-KD into existing KD pipelines and evaluate it extensively across classification, instruction-following, and summarization tasks (13 datasets, 8 model pairs).Flex-KD consistently improves student performance over baselines (up to 3.75% ROUGE-L gain) while remaining stable and computationally efficient

**Strengths:**

1.Novel and elegant idea – The parameter-free subspace selection via gradient attribution is both conceptually simple and technically effective. It directly addresses a longstanding limitation in feature-based KD for LLMs.
2.Strong empirical validation – The experiments span a wide range of architectures (GPT2, BERT, OPT, LLaMA, BART) and tasks. The results are consistent and show meaningful improvements over strong baselines such as Projector-based and CKA-based feature KD.
3.Efficiency and practicality – Flex-KD adds no trainable parameters, integrates seamlessly into existing KD pipelines, and performs robustly even under limited training data (5% of Dolly).

**Weaknesses:**

Limited theoretical justification – The use of gradient magnitude as a proxy for task relevance is intuitive but lacks a rigorous theoretical foundation or ablation on its correlation with mutual information or task-specific utility.
2.Computation cost concern – Gradient-based importance estimation still requires additional backward passes over the dataset, which may become expensive for extremely large teacher models. Efficiency comparisons are missing.
3.Scope of applicability – Experiments are limited to Transformer-based architectures; it remains unclear whether Flex-KD can generalize to non-Transformer or multi-modal settings (e.g., Mamba, vision-language models).

**Questions:**

1.How often are the gradient-based importance scores recomputed during training? Is the subspace selection static (computed once) or dynamic (updated per epoch)?
2.What is the computational overhead of Flex-KD compared to projector-based distillation?
3.Could the proposed method be combined with structured pruning or neuron masking for further compression?
4.Would gradient saturation in very deep networks affect the stability of importance estimation?
5.How does Flex-KD behave when the student’s hidden size is larger than the teacher’s (e.g., upscaling scenarios)?

---

> ### Author Response · Authors · 2025-11-28
> **Response to Reviewer HhK4 (1/2)**
>
> We sincerely appreciate your review. Blow, we address your concerns point by point.
>
> >**Q1:** Limited theoretical justification – The use of gradient magnitude as a proxy for task relevance is intuitive but lacks a rigorous theoretical foundation or ablation on its correlation with mutual information or task-specific utility.
>
> Identifying task-relevant neurons via gradient-based attribution is widely used in recent LLM work (Iurada et al., 2025; Muralidharan et al., 2024). Importantly, we do not rely on this choice without empirical validation. First,**in Figure 3a of section 4.1,** we compare three frequently used attribution methods, standard gradients, activation magnitudes, and Integrated Gradients (IG), and observe that standard gradients consistently yield the best downstream performance with the lowest variance. Second, **in response to the reviewer’s request, we provide an additional ablation**, in the following Table, in which we systematically remove the top-ranked, lowest-ranked, or random units from the teacher across 3 GLU dataset ( RTE, STS-B, and SST-2). Removing top-ranked units (high gradient score) causes dramatic performance degradation (e.g., $-13\%$ on RTE, $-6.5\%$ on STSB)
> whereas removing low-ranked units has negligible effect. This strongly indicates that gradient magnitude correlates with true task utility, not merely local sensitivity. Together, these experiments offer concrete empirical justification that gradient-based importance is a reliable signal for identifying task-relevant teacher subspaces.
>
> **Table**: Ablation study across RTE, STS-B, and SST-2. For STS-B, we report the average of Pearson and Spearman.
>
> | Ablation Type     | % Removed | RTE Acc | Δ Acc | STS-B Avg | Δ Avg | SST-2 Acc | Δ Acc |
> |-------------------|-----------|---------|-------|-----------|--------|-----------|--------|
> | **High Importance** | 0%        | 68.23   | –     | 88.23     | –      | 94.49     | –      |
> |                   | 10%       | 60.29   | −7.9  | 87.09     | −1.1   | 94.27     | −0.2   |
> |                   | 20%       | 55.23   | −13.0| 87.09 | −1.1   | 94.15     | −0.3   |
> |                   | 50%       | 54.51 | **−13.7** | 81.75 | **−6.5** | 93.46 | **−1.0** |
> | **Low Importance**  | 0%        | 68.23   | –     | 88.23     | –      | 94.49     | –      |
> |                   | 10%       | 68.59   | +0.4  | 88.23     | 0.0    | 94.50     | 0.0    |
> |                   | 20%       | 68.23   | 0.0   | 88.21     | 0.0    | 94.50     | 0.0    |
> |                   | 50%       | 68.59   | +0.4  | 88.23     | 0.0    | 94.15     | −0.3   |
> | **Random**          | 0%        | 68.23   | –     | 88.23     | –      | 94.49     | –      |
> |                   | 10%       | 68.95   | +0.7  | 88.22     | 0.0    | 94.38     | −0.1   |
> |                   | 20%       | 69.68   | +1.4  | 88.12     | −0.1   | 93.92     | −0.6   |
> |                   | 50%       | 66.43   | −1.8  | 87.96     | −0.3   | 94.27     | −0.2   |
>
> >**Q2&Q3:** Computation cost concern – Gradient-based importance estimation still requires additional
> backward passes over the dataset. Efficiency comparisons are missing
>
> We emphasize that Flex-KD computes gradients only with respect to the **teacher’s final hidden layer**, not through all layers. Consequently, the cost of importance estimation is independent of the teacher’s depth and remains lightweight even for large models. We emphasize also that we compute the gradient **only one time before training the student**. This one-time gradient computation requires a single backward pass over the dataset and takes only a few seconds on our tasks, as reported in Table 9 (on XSum and CNN datasets).
>
> During training, Flex-KD adds no substantial overhead: the feature-alignment loss operates solely on the selected subspace of the last layer, making its cost negligible relative to the forward/backward passes already required for logit KD. **To address efficiency concerns directly**, we additionally measured full training time (hours per epoch) for pure logit KD (KD) and for Projector, CKA-KD, and Flex-KD. As shown below, Flex-KD is the most efficient feature-distillation method, substantially faster than both Projector and CKA, because it avoids full-layer matching and performs alignment **only at the final encoder/decoder layers**. These results confirm that Flex-KD introduces minimal overhead while achieving significantly stronger performance.
>
> **Table**: Computational cost (hours/epoch) of different feature distillation methods
> (6×640 BART student, single A100 GPU, XSum dataset).
>
> | **Method**                                | **Hours / Epoch** |
> |-------------------------------------------|--------------------|
> | KD (Hinton et al., 2015)                  | 0.93               |
> | Projector (Jiao et al., 2020)             | 1.94               |
> | CKA-KD (Dasgupta et al., 2025)            | 1.35               |
> | **Flex-KD (ours)**                        | **0.93**           |

---

> ### Author Response · Authors · 2025-11-28
> **Response to Reviewer HhK4 (2/2)**
>
> >**Q5:** How often are the gradient-based importance scores recomputed during training? Is the subspace selection static (computed once) or dynamic (updated per epoch)?
>
> The gradient-based importance scores are computed once before training, and the selected teacher subspace remains static throughout distillation. In fact, in Figure 3a of subsection 4.1, We conduct an ablation study comparing static selection with dynamic selection (recomputing scores per batch or per epoch). Dynamic selection introduced instability and led to substantial performance drops (up to 8\%), whereas static selection was both more stable and more accurate.
>
> >**Q6:** What is the computational overhead of Flex-KD compared to projector-based distillation?
>
> Please refer to Q2&Q3 answer.
>
> >**Q4&Q7:**  Experiments are limited to Transformer-based architectures (e.g., Mamba, vision-language models). Could the proposed method be combined with structured pruning or neuron masking for further compression?
>
> Thank you for raising this point. In this work, we deliberately focused on providing a thorough and controlled evaluation of Flex-KD against competing methods across diverse settings: classification, summarization, instruction following, and low-data regimes, using multiple backbone architectures to isolate and assess the core contribution of task-driven with different hidden size feature distillation.
>
> While combining Flex-KD with structured pruning or using  non-Transformer is indeed an interesting direction, it falls outside the scope of our current study. Prior work has shown that pruning and KD can be complementary. Hence, we believe integrating Flex-KD into such pipelines is a promising avenue for future research.
>
> >**Q8:** Would gradient saturation in very deep networks affect the stability of importance estimation?
>
> Again, we emphasize that Flex-KD computes gradients **only with respect to the teacher’s final hidden layer**, not through the full network. Therefore, it does not suffer from gradient saturation issues typically associated with deep backpropagation, and the stability of importance estimation is not affected by model depth. In fact, we empirically verified the robustness of our attribution mechanism. **In Figure 7 (Appendix), we evaluate the consistency of the selected units across five random seeds**. The resulting importance rankings exhibit very high overlap, 91.5\% on XSum and 96.3\% on CNN/DailyMail, indicating that our gradient-based selection is both stable and reliable.
>
> >**Q9:** How does Flex-KD behave when the student’s hidden size is larger than the teacher’s (e.g., upscaling scenarios)?
>
> Flex-KD is designed for the standard knowledge distillation setting, where knowledge from a larger teacher model is transferred to a smaller student for model compression. In this regime, the student naturally has a smaller hidden size than the teacher. The purpose of Flex-KD is precisely to remove the architectural constraint that previously required matching hidden dimensions between teacher and student, thereby enabling more flexible, and often stronger compression ratios.
> Upscaling (i.e., using a student larger than the teacher) is not a typical objective in KD for compression and falls outside the scope of our method. Flex-KD is therefore not intended for such inverse scenarios.
>
> We hope this clarification addresses your concerns, and we kindly invite the reviewer to reconsider the evaluation in light of these additional clarifications and results.

---

### Meta-Review · Area_Chair_6af8 · 2026-01-06

**Summary:**

The concerns of reviewers mainly focus on following aspects.

Theoretical and methodological design: Using gradient magnitude as a task relevance metric lacks a rigorous theoretical foundation, and its correlation with task utility has not been fully verified; the definition of "task relevance" may ignore nonlinear interactions between hidden units; only global static feature selection is adopted, and input-dependent dynamic selection strategies are not explored, leading to limited applicability. Overall, the article lacks sufficient theoretical innovations and fails to propose a relatively innovative distillation paradigm.

Experiment and comparison completeness: The coverage of baseline methods is insufficient, with no comparison with the latest logit-based KD methods such as GKD and DistiLLM-2; experiments are limited to teacher-student models from the same family, and the feasibility of cross-family distillation has not been verified; some reviewers believe that the experimental models are outdated, tasks are simple, comparison methods are obsolete, and the advantages are not significant; the performance of hybrid strategies with logit-based KD has not been verified.

Computational efficiency: Gradient calculation requires additional backpropagation, and the computational cost for ultra-large teacher models is questionable; the overhead comparison with lightweight logit-based KD is not clear, and the additional cost is ambiguous.
Extended applicability: Experiments only cover the Transformer architecture, and the generality in non-Transformer (e.g., Mamba) or multimodal scenarios has not been verified; the combined compression potential with structured pruning and neuron masking has not been explored; it does not support the "upgrade" scenario where the hidden layer size of the student model is larger than that of the teacher.

Given the issues discussed above, the paper does not appear to fully meet the standard for acceptance.

**Reviewer Concerns:**

Comments effectively addressed through rebuttals

(1)	Theoretical support-related: The authors verified the strong correlation between gradient magnitude and task utility through ablation experiments (significant performance degradation when removing high-gradient units) and cited recent LLM studies to prove the rationality of gradient attribution; experiments comparing static and dynamic selection confirmed that static selection is more stable and achieves better performance.

(2)	Experiment and comparison-related: Supplementary comparative experiments with the latest baselines such as GKD and DistiLLM show that Flex-KD achieves performance improvements on their basis; it is clarified that the compared Projector and CKA-KD are directly relevant SOTA methods for solving "hidden layer size mismatch", refuting the doubt that "comparison methods are outdated"; it is explained that the hybrid strategy with logit-based KD is a conventional design in the field, and experiments prove that the hybrid strategy only increases a small amount of one-time cost and achieves better performance.

(3)	Computational efficiency-related: It is clarified that Flex-KD only requires one gradient calculation of the teacher's final hidden layer before training (taking a few seconds), with no additional high overhead during training; experimental data is provided to prove that its training time is equivalent to pure logit-based KD and significantly lower than that of Projector and CKA-KD.

(4)	Rationality of models and tasks-related: It is emphasized that the experiments adopt mainstream models such as GPT and LLaMA (with teachers up to 6.7B parameters), which are consistent with the experimental scale of recent top conference papers, refuting the evaluation of "using toy models"; supplementary experiments on large instruction-following datasets such as S-NI and UnNI verify the effectiveness of the method on complex tasks.

Outstanding comments not yet addressed

(1)	Insufficient theoretical depth: Although the effectiveness of gradient magnitude has been verified through experiments, rigorous theoretical derivation is still lacking, and its correlation with indicators such as mutual information has not been clarified.

(2)	Limitations in extended applicability: Experimental verification in non-Transformer architectures and multimodal scenarios has not been supplemented; the combined compression effect with structured pruning has not been explored; it is clearly stated that the scenario where "the hidden layer size of the student model is larger than that of the teacher model" is not supported, and the potential needs of this scenario have not been addressed.

(3)	Feasibility of cross-family distillation: The authors point out that the core difficulty of cross-family distillation is tokenizer mismatch, but no solutions or preliminary verifications are provided, and this issue remains unclear.

(4)	Further exploration of dynamic selection strategies: Although it is confirmed that static selection is better, more flexible compromise schemes for "input-dependent tasks" have not been explored, and only the effectiveness of dynamic selection is denied.

**Reviewer Scores:**

(1)	Reviewer HhK4

Initial score: 2

Predicted score adjustment: Adjusted to 4

Reasons for adjustment: The initial score is inconsistent with the sub-item evaluation of "soundness, presentation, contribution are all good"; the authors have provided sufficient experimental data and detailed explanations for all the issues raised regarding theoretical support, computational cost, applicability, etc., and the core concerns have been addressed, but the problem of insufficient theoretical depth has not been fully compensated, so direct acceptance is not supported.

(2)Reviewer X9ZQ

Initial score: 4

Predicted score adjustment: 4

Reasons for adjustment: The authors clarified the essential difference between Flex-KD and CKA-KD through experiments (selective alignment vs. undifferentiated alignment), confirming that it is not incremental innovation; supplementary overhead comparison data with logit-based KD addresses the concern about computational cost; regarding the question of dynamic selection, the conclusion that static selection is better is verified through experiments, and all core concerns have been effectively responded to.

(3)Reviewer uBcB

Initial score: 0

Predicted score adjustment: Adjusted to 4

Reasons for adjustment: The authors fully refuted the evaluations of "using toy models" and "comparison methods are outdated", clarified that the scale and settings of models such as GPT and LLaMA used in experiments comply with the mainstream research norms in the field, and supplemented comparative experiments with the latest baselines; however, the comment that "the definition of task relevance does not consider nonlinear interactions between hidden units" has not been fully addressed, so it still does not meet the standard of full acceptance.

(4)Reviewer GbPn

Initial score: 6

Predicted score adjustment: 6

Reasons for adjustment: The authors supplemented comparative experiments with the latest logit-based KD baselines and verified the performance of the hybrid strategy between Flex-KD and such methods, and the core concerns have been addressed; although the feasibility of cross-family distillation has not been verified, this issue is an extensible requirement and not an essential part of the core contribution of the method, which does not affect the recognition of core effectiveness, so the original score is maintained.

---

### Decision · Program_Chairs · 2026-01-26

Reject